# Primary cilia regulate hematopoietic stem and progenitor cell specification through Notch signaling in zebrafish

Zhibin Liu[1,2,3], Haiqing Tu[4], Yunsi Kang[5], Yuanyuan Xue[1,2,3], Dongyuan Ma[1,2,3], Chengtian Zhao [5], Huiyan Li[4], Lu Wang[1,2,6] & Feng Liu [1,2,3]

Hematopoietic stem and progenitor cells (HSPCs) are capable of producing all mature blood lineages, as well as maintaining the self-renewal ability throughout life. The hairy-like organelle, cilium, is present in most types of vertebrate cells, and plays important roles in various biological processes. However, it is unclear whether and how cilia regulate HSPC development in vertebrates. Here, we show that cilia-specific genes, involved in primary cilia formation and function, are required for HSPC development, especially in hemogenic endothelium (HE) specification in zebrafish embryos. Blocking primary cilia formation or function by genetic or chemical manipulations impairs HSPC development. Mechanistically, we uncover that primary cilia in endothelial cells transduce Notch signal to the earliest HE for proper HSPC specification during embryogenesis. Altogether, our findings reveal a pivotal role of endothelial primary cilia in HSPC development, and may shed lights into in vitro directed differentiation of HSPCs.

[1] State Key Laboratory of Membrane Biology, Institute of Zoology, Chinese Academy of Sciences, 100101 Beijing, China. [2] University of Chinese Academy of Sciences, 100049 Beijing, China. [3] Institute for Stem Cell and Regeneration, Chinese Academy of Sciences, 100101 Beijing, China. [4] State Key Laboratory of Proteomics, National Center of Biomedical Analysis, 100850 Beijing, China. [5] Institute of Evolution and Marine Biodiversity, Ocean University of China, 266003 Qingdao, China. [6] State Key Laboratory of Experimental Hematology, Institute of Hematology and Blood Diseases Hospital, Chinese Academy of Medical Sciences & Peking Union Medical College, 300020 Tianjin, China. Correspondence and requests for materials should be addressed to L.W. (email: wanglu1@ihcams.ac.cn) or to F.L. (email: liuf@ioz.ac.cn)

I t has been well established that most vertebrate cells can transmit extracellular signals through a hairy-like sensory organelle, called primary cilium[1]. Primary cilia exist in many types of cells including endothelial cells (ECs), epithelia, fibroblasts, and others in vertebrates[2]. Previous studies have revealed that primary cilia have specialized functions, in shear-stress sensation[3,4], chemosensation[5], differentiation[6], proliferation[7], and maintenance of stem cells in a wide array of tissues[2,8–11]. Thus, defects in ciliogenesis and function usually lead to ciliopathies, such as autosomal dominant polycystic kidney disease, obesity, and others[10].

In the cilia system, primary cilia contain a "9 + 0" axoneme ("9" denotes nine parallel doublet microtubules and "0" denotes absence of a central pair of microtubules (MTs)). An intraflagellar transport (IFT) system, including intraflagellar transport protein-88 (IFT88 or Polaris), is utilized to elongate MT axoneme. Ift88 knockdown (KD) led to a vascular impairment phenotype[12]. Furthermore, a calcium channel protein, Polycystin1 (PKD1), is localized in cilia, and PKD1 conventional knockout mice are embryonically lethal at E15.5 due to vascular leakages and hemorrhage[13]. Similarly, Polycystin2 (PKD2) KD also caused angiogenesis defects in zebrafish[12]. Interestingly, a recent study demonstrated that primary cilia are present in the ECs of zebrafish blood vessels[12]. Given that a subset of ECs in the dorsal aorta (DA) can develop into hemogenic endothelial cells (HE cells), it is tempting to speculate that the endothelial primary cilia in the DA may participate in HE specification.

During hematopoietic stem and progenitor cell (HSPC) development, HE cells produce HSPCs through the endothelial-to-hematopoietic transition (EHT) in vertebrates[14–16]. Nonetheless, our understanding of the precise regulatory mechanisms involved in HE specification is still limited[17]. The transcription factor runx1 is a widely used marker for HE cells at the early embryonic stage[18]. Deficiency of Runx1 results in impairments of EHT and definitive hematopoiesis[19,20]. Notch signaling is a critical regulator of runx1. In vertebrates, loss of one of Notch receptors, Notch1, causes a decrease of runx1 expression, which subsequently affects definitive hematopoiesis[21–23]. Furthermore, in the absence of Notch ligands, the definitive hematopoiesis is also disrupted in jagged1 null mice[24] and zebrafish mindbomb mutants[25]. Notch signaling exerts complex regulation in HSPC development through divergent ligands and receptors[26,27], as well as multiple inputs[28,29]. However, very little is known about the upstream factors of Notch signaling and how they initiate Notch activation. Intriguingly, it has been reported that Notch components localize in cilia and Notch signaling can be transmitted through cilia[30,31]. However, it remains elusive whether cilia can transduce Notch signaling in controlling definitive hematopoiesis in vertebrates.

Here, we use the zebrafish as a vertebrate model and demonstrate that impairment of primary cilia formation or function leads to defects in HSPC development, especially in HE specification. Blocking primary cilia specifically in ECs causes the reduction of HE cells. Mechanistically, we uncover that Notch signaling functions downstream of endothelial primary cilia to specify HE cells properly. Altogether, our findings demonstrate that endothelial primary cilia modulate HSPC development through transducing Notch signaling.

## Results

### The dynamics of endothelial cilia during embryogenesis.
To study the underlying link between cilia and hematopoiesis, primary cilia in the vascular ECs in the aorta-gonad-mesonephros (AGM) region, where the definitive hematopoiesis occurs, were characterized firstly. By visualizing a triple-transgenic line, Tg (βact:Arl13b–GFP/kdrl:mCherry/runx1:en-GFP), which marks cilia, ECs and hematopoietic cells (including HSPCs), respectively, we found that primary cilia were present in ECs in the AGM region at 28 h post fertilization (hpf) (Fig. 1a). Meanwhile, $runx1^+kdrl^+$ HE cells were also ciliated (Fig. 1b), which was supported by analysis of another HE transgenic line, Tg(gfi1:GFP/ βact:Arl13b–GFP), at 28 hpf (Fig. 1b). In contrast, the cmyb-labeled HSPCs in the AGM region were non-ciliated (Ac-tubulin labeled cilia) by fluorescence in situ hybridization (FISH) and Ac-tubulin staining (Fig. 1c). Time-course analysis of a double-transgenic line, Tg(βact:Arl13b–GFP/ kdrl:mCherry) showed that the number of primary cilia was reduced from 32 hpf and nearly absent at 52 hpf in the AGM region (Fig. 1d–f). The dynamic changes of primary cilia were consistent with previous observation of cilia in the caudal artery and caudal vein[12]. Interestingly, the dynamic changes of endothelial primary cilia occurred between 24 and 52 hpf, which is the time window critical for HSPC emergence, indicating a possible relationship between HSPC development and primary cilia localized in ECs.

### Identification of genes that regulate endothelial cilia.
To functionally investigate the requirements of critical genes in ciliogenesis, we first performed KD experiments using antisense morpholino oligonucleotides (MOs) against cilia genes pkd2, kif3a, and ift88, which have been shown to be involved in cilia formation and function[32–34], and a cilia gene fsd1, which is required for ciliogenesis[35]. The efficiency of fsd1 aMO (translation blocking morpholino) and sMO (splice morpholino) was validated by western blotting and RT-PCR, respectively[35]. The specificity of pkd2, kif3a, and ift88 MOs was validated by GFP reporter assay. The EGFP expression was blocked by co-injection of corresponding MOs at one-cell stage, respectively (Supplementary Fig. 1A). As loss of cilia genes usually causes defects in the left-right asymmetry and body-curvature phenotypes in vertebrates[36,37], we first noticed abnormal spaw expression in the lateral plate mesoderm at 18-somite stage, as well as disordered heart looping upon fsd1 KD[35]. In addition, body curvature was observed in the other three types of cilia-impaired embryos (Supplementary Fig. 1B and 1C), consistent with previous reports[32–34]. The three-dimension (3D) ultrastructure of both endothelial primary cilia in the AGM region and motile cilia in pronephric duct (PD) were characterized by transmission electron microscope (TEM). A canonical "9 + 0" axoneme was observed in the AGM region and a canonical "9 + 2" axoneme was observed in the PD in both control and fsd1 morphants, indicating that the 3D ultrastructure of primary cilia was unaltered (Fig. 2a). Live confocal imaging of endothelial primary cilia in these morphants was performed using Tg(βact:Arl13b–GFP/ kdrl:mCherry) double-transgenic line to further examine the cilia phenotype. The results showed that both the number and length of cilia were affected in fsd1 morphants (Fig. 2b–d and Supplementary Fig. 2A–F), as well as in pkd2, kif3a and ift88 morphants (Fig. 2e, f), compared to controls. The above results demonstrate that cilia genes fsd1, pkd2, kif3a, and ift88 are indeed required for ciliogenesis in zebrafish embryos.

### Ciliogenesis is essential for HSPC development.
As primary cilia are present in ECs in the AGM region (Fig. 1a, d–f), from where the earliest HSPCs are derived, our quantitative RT-PCR (qPCR) analysis showed that fsd1 was highly enriched in $kdrl^+runx1^+$ cells (HE cells; Supplementary Fig. 3A). These data together indicated a potential role of cilia in hematopoiesis. To test this hypothesis, we first detected the expression level of HSPC markers, runx1 and cmyb, in cilia-impaired embryos. The decreased expression of HSPC markers was observed in fsd1 morphants at

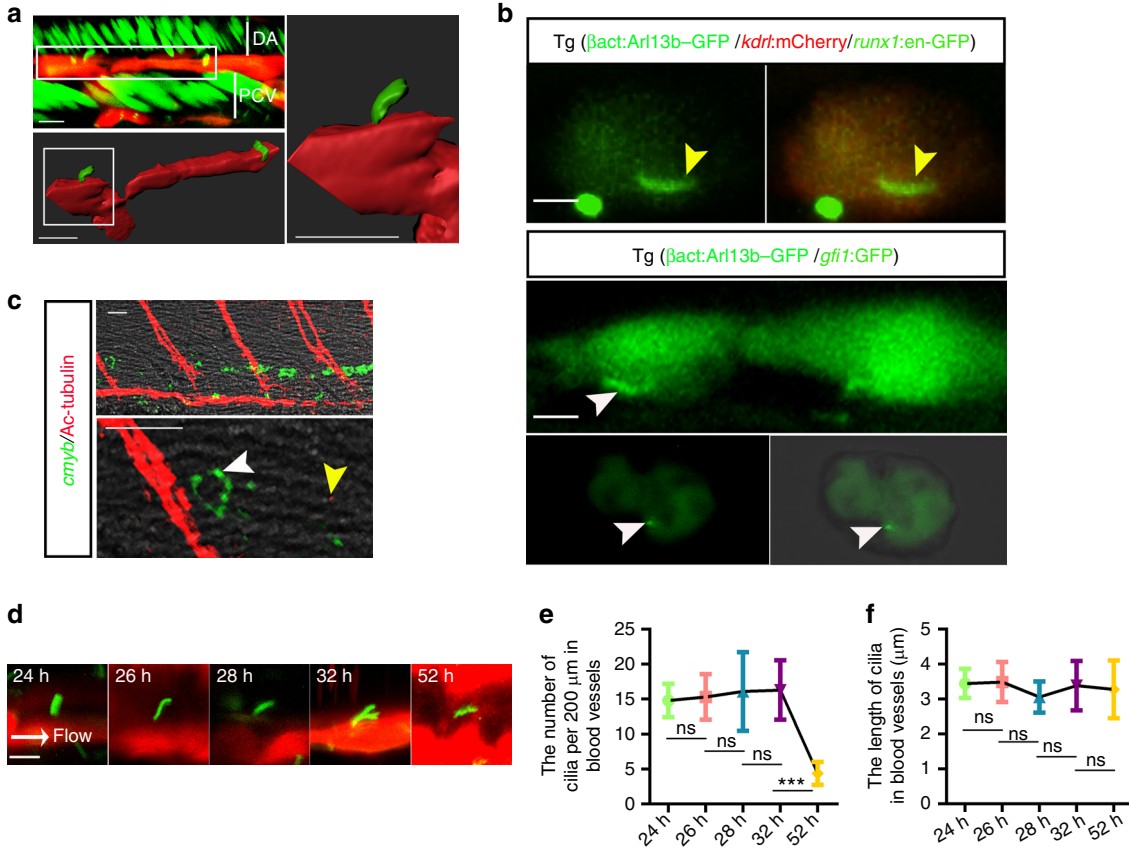

**Fig. 1** Ciliogenesis occurs in vascular endothelial cells (ECs) in AGM. **a** Three-dimension (3D) confocal imaging showing cilia on ECs in the AGM region using Tg(βact:Arl13b–GFP/*kdrl*:mCherry/*runx1*:en-GFP) line at 28 hpf. White squares indicate the ECs with cilia. White bars denote DA or PCV region. DA dorsal aorta, PCV posterior cardinal vein. Scale bars, 10 μm. **b** The live confocal imaging of $kdrl^+$/$runx1^+$ or $gfi1^+$ HE cells in Tg(βact:Arl13b–GFP/*kdrl*: mCherry/*runx1*:en-GFP) (upper panel) or Tg(βact:Arl13b–GFP/*gfi1*:GFP) at 28 hpf (middle panel). The imaging of $gfi1^+$ HE cells sorted by fluorescence-activated cell sorting of dissected trunk region in Tg (*gfi1*:GFP/βact:Arl13b–GFP) embryos (lower panel). Yellow arrowheads indicate primary cilia in $kdrl^+$/$runx1^+$ cells; white arrowheads indicate primary cilia in $gfi1^+$ HE cells. Scale bars, 5 μm. **c** Fluorescence in situ hybridization (FISH) result showing the *cmyb* expression and Ac-tubulin staining showing the cilia in the aorta-gonad-mesonephros (AGM) region at 48 hpf. Yellow arrowhead indicates primary cilia and the white arrowhead indicates the *cmyb*$^+$ HSPC in the AGM region. The *cmyb* probe was used to examine *cmyb* expression in Tg (*cmyb*:EGFP) embryos by FISH. Scale bars, 20 μm. **d** The live confocal imaging of cilia with *kdrl*:mCherry/βact:Arl13b–GFP double-transgenic line. White arrow denotes the blood flow direction. Scale bar, 5 μm. **e, f** The statistical data shows the primary cilia number (**e**) and length (**f**) in blood vessels in the AGM region in wild-type embryos. The cilia length presented in each embryo was the average length of all the cilia in the DA of the AGM region calculated per 200 μm. Data represent the analysis results of one-way ANOVA–Sidak test. Error bars, mean ± s.d., $n = 10$ embryos. ns non-significant, ***$P < 0.001$

36 hpf, but not in *fsd1* mismatch morpholino (misMO)-injected embryos (Fig. 3a and Supplementary Fig. 3B). Consistently, the protein level of Runx1 was also decreased (Fig. 3b).

Furthermore, we generated an *fsd1* mutant[35] by CRISPR/Cas9 (with a 20-base-pair insertion in the 4th exon of *fsd1*, Supplementary Fig. 3C) and the reduced expression of *runx1* and *cmyb* in *fsd1* mutants (maternal-zygotic mutants, $fsd1^{-/-}$) mimicked the defects in *fsd1* morphants (Fig. 3c, d). To further verify that the HSPC defects were indeed specific to *fsd1*, we performed rescue experiments by overexpression of human full-length *FSD1* mRNA (FSD1 hmRNA, escaping from *fsd1* aMO targeting) in *fsd1* morphants at one-cell stage. According to whole-mount in situ hybridization (WISH) results, we observed restoration of *runx1* expression in FSD1 hmRNA-injected *fsd1* morphants, compared to *fsd1* morphants alone (Supplementary Fig. 4A).

To investigate whether *fsd1* specifically in vascular ECs influences HSPCs, we constructed a plasmid in which the expression of *fsd1* fused with an EGFP reporter was driven by *fli1a* promoter (fli1a:fsd1-EGFP). Overexpression of *fsd1* specifically in ECs could rescue the HSPC defects in *fsd1* morphants

(Supplementary Fig. 4B and 4C), suggesting that the observed HSPC defects were specific to *fsd1* in ECs.

In addition, the other three cilia-dysfunction embryos (*pkd2*, *kif3a*, and *ift88* morphants) showed similar HSPC defects (Fig. 3e). Furthermore, we injected a sub-effective dose of *pkd2* (0.5 ng) or *ift88* (0.8 ng) morpholinos, which did not affect HSPC development in wild-type embryos into the *pkd2* or *ift88* mutants[38] to KD the residual mRNA of *pkd2* or *ift88*, respectively (Supplementary Fig. 5A–F). As expected, we observed the decreased expression of *runx1* at 36 hpf, compared to controls (Fig. 3f), suggesting that these ciliary genes are indeed required for definitive hematopoiesis.

To further demonstrate the role of cilia in definitive hematopoiesis, ciliobrevin D (CBD) treatment was performed. CBD is a AAA + ATPase motor cytoplasmic dynein inhibitor and is used to block cilia function[39,40]. Wild-type embryos were treated with CBD from 10 to 31 hpf, and the results showed that cilia were unaltered (Fig. 4a–c). However, the expression of Notch target genes was reduced (Supplementary Fig. 6A), which suggests that CBD treatment is effective as previously reported[40]. WISH data showed that expression of *runx1* in CBD-treated

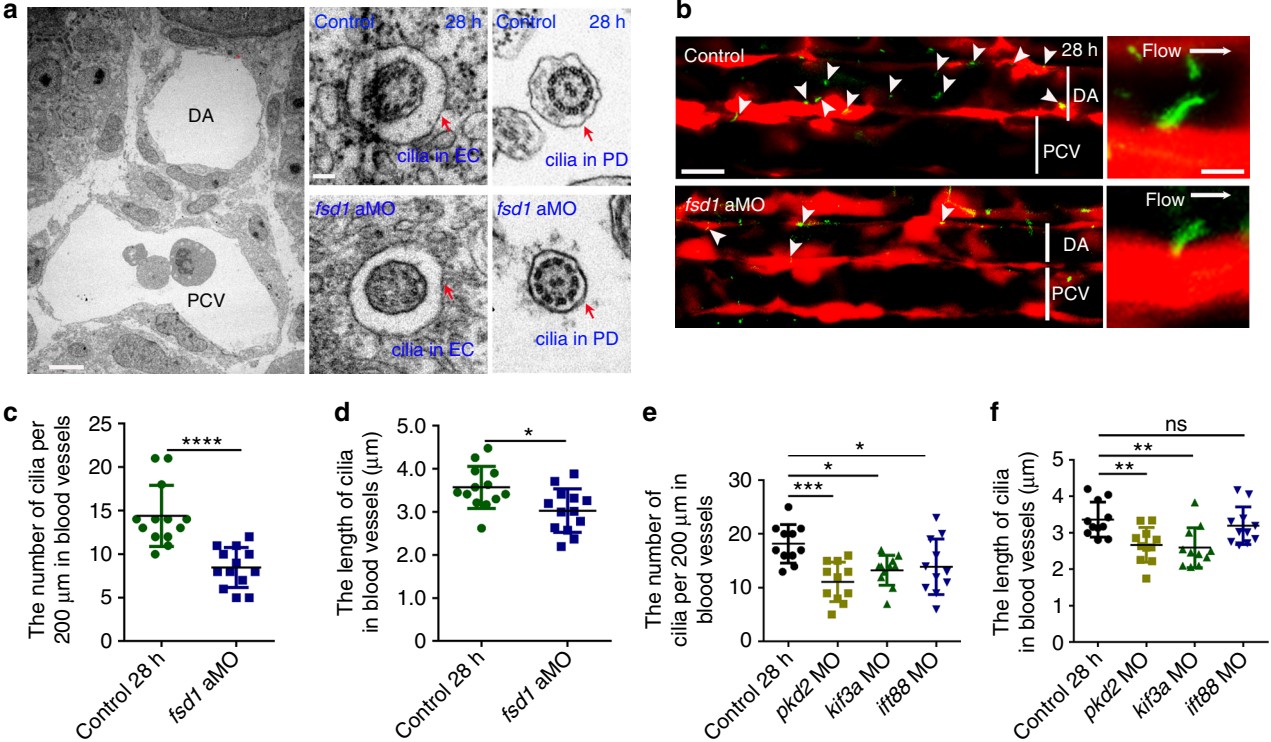

**Fig. 2** Loss of cilia genes causes primary cilia defects in blood vessels in the aorta-gonad-mesonephros (AGM) region. **a** Transmission electron microscopy (TEM) imaging of blood vessels (left panel) in the AGM region in control and *fsd1*-deficient embryos at 28 hpf. The red arrows indicate the ultrastructure of primary cilia in vascular endothelial cells (ECs) (middle panels) and motile cilia in pronephric duct (right panels). Red arrows denote cilia. Scale bars, 5 μm (left panel) and 0.1 μm (right panel). **b** Visualization of cilia in ECs in the AGM region using *kdrl*:mCherry/βact:Arl13b–GFP double-transgenic line in control and *fsd1* morphants at 28 hpf. The white arrowheads indicate the primary cilia in blood vessels. White bars denote DA or PCV region. Scale bars, 20 μm (left panel) and 5 μm (right panel). **c–f** The quantification of the primary cilia number and length with *kdrl*:mCherry/βact:Arl13b–GFP double-transgenic line in control and cilia-impaired embryos at 28 hpf. Data in **c**, **d** were analyzed by Student's *t*-test (*n* = 13 embryos). Data in **e**, **f** represent the analysis results of one-way ANOVA–Dunnett test (*n* = 11 embryos). Error bars, mean ± s.d. ns non-significant, *$P < 0.05$, **$P < 0.01$, ***$P < 0.001$, ****$P < 0.0001$

embryos was decreased compared with control group (Fig. 4d). These results imply that cilia are required for HSPC formation.

To gain further evidence that primary cilia in ECs affect HSPC development, impairment of cilia specifically on ECs of blood vessels was achieved by injection of the CRISPR vector *fli1a:ift88*-cKO (to genetically delete *ift88* specifically in ECs) together with Tol2 mRNA (Supplementary Fig. 6B–D)[40]. Consistent with CBD treatment, the live confocal imaging data showed similar cilia defects (Fig. 4a–c) and the number of HE cells was decreased in endothelial primary cilia-dysfunction embryos (Fig. 4e). Collectively, these results support that cilia in ECs are essential for HSPC development.

**Cilia are required for HE specification.** To examine when and how exactly HSPC defects occurred in cilia-dysfunction embryos, we next focused on *fsd1*-deficient embryos. Primitive hematopoietic markers were examined by WISH at 24 hpf and their expression was relatively normal in *fsd1* morphants (Supplementary Fig. 7), indicating that *fsd1* is dispensable for primitive hematopoiesis. To determine the exact role of *fsd1* in definitive hematopoiesis, we examined the time-course expression of *runx1* and found that the *runx1* expression was decreased from 26 hpf in *fsd1* morphants, when HE specification normally occurred in control embryos (Fig. 5a). To confirm the phenotype mentioned above, we examined the other two HE markers, *gfi1aa* and *gata2b*. The expression of *gfi1aa* and *gata2b* was also decreased in *fsd1* morphants at 26 hpf (Fig. 5a), suggesting that HE specification was disrupted in the

absence of *fsd1*. Furthermore, HSPC derivatives such as erythroid cells (*gata1*), myeloid cells (*pu.1*) in the CHT, and T cells (*rag1*) in the thymus were all decreased at 4 days post fertilization (dpf) in *fsd1* morphants (Fig. 5b). The defects of HE specification as well as HSPC derivatives were also observed in *fsd1* sMO-injected embryos and *fsd1* mutants (Supplementary Fig. 3B and 8; Figs. 3c and 5c). Moreover, the number of HE cells was quantified by counting the *kdrl*:mCherry[+]/*runx1*:en-GFP[+] or *kdrl*:mCherry[+]/*cmyb*:EGFP[+] double positive cells in the AGM region at different developmental stages. In *fsd1* morphants, the numbers of HE cells were significantly reduced compared to controls (Fig. 5d–g). Thus, these results support that *fsd1* is required for HE specification. Similarly, loss-of-function of *pkd2*, *kif3a*, or *ift88* and in particular, conditional deletion of *ift88* in vascular ECs, all led to impaired HE specification (Fig. 5h–l). Taken together, we conclude that cilia defects in vascular ECs impair HE specification.

Given that HSPCs are originated from the DA and the vessel integrity/identity is essential for HSPC emergence, we next quantified the EC number within DA in the AGM region by using Tg(*kdrl*:mCherry/*fli1a*:nGFP) transgenic line (Supplementary Fig. 9A). In addition, expression of arterial markers *dll4*, *ephrinB2a*, venous markers *msr*, *flt4*, and also their niche cell markers *kdrl* (pan-EC), *myod* (somite cell), and *pax2.1* (PD) was examined (Supplementary Fig. 9B and 9C). The results showed that both the EC numbers in DA and expression of vessel marker genes were not altered in *fsd1* morphants. Similarly, normal blood vessels were observed in the other three cilia defective embryos (Supplementary Fig. 9D–F), albeit with a slightly curved body

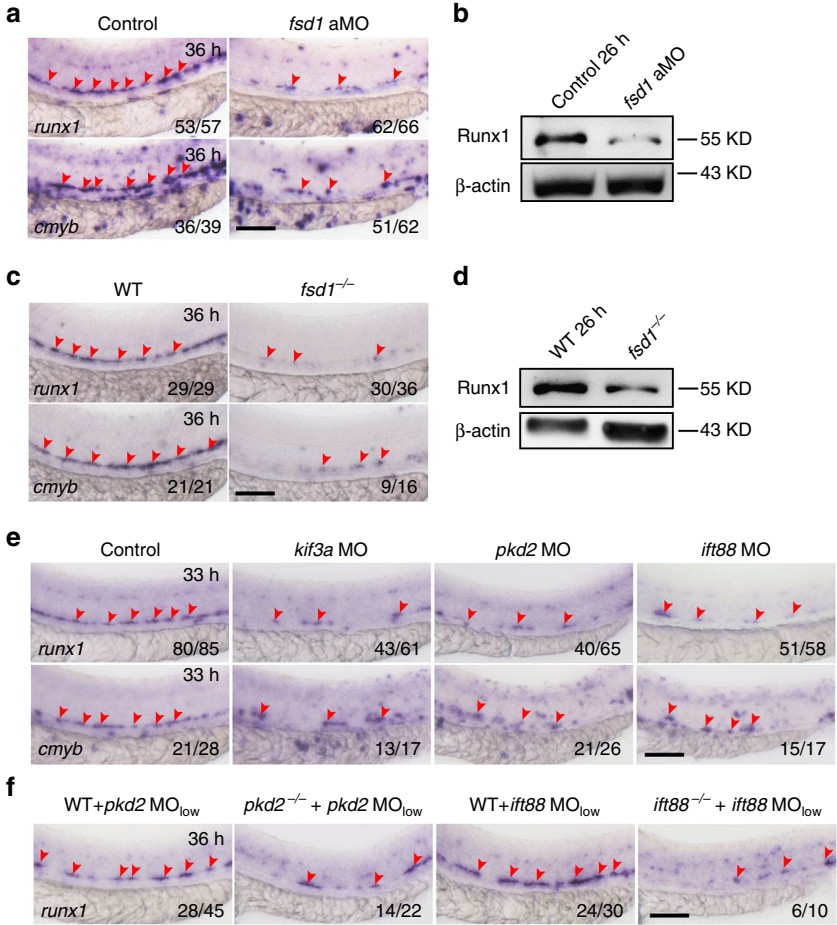

**Fig. 3** Loss of cilia genes induces hematopoietic stem and progenitor cell (HSPC) defects. **a** Whole-mount in situ hybridization (WISH) results of HSPC markers (*runx1* and *cmyb*) in the aorta-gonad-mesonephros (AGM) region in control and *fsd1* morphants at 36 hpf. The red arrowheads indicate the expression of HSPC markers *runx1* and *cmyb* in DA. **b** Western blotting showing the protein level of Runx1 in control and *fsd1* morphants at 26 hpf. **c** WISH analysis showing the expression of HSPC markers *runx1* and *cmyb* (red arrowheads) in the DA in *fsd1*[−/−]. **d** Western blotting showing the protein level of Runx1 in *fsd1*[−/−]. **e** Representative images showing *runx1* and *cmyb* expression in control, *pkd2*, *kif3a*, and *ift88* morphants at 33 hpf. The red arrowheads indicate the expression of HSPC markers *runx1* and *cmyb*. **f** WISH analysis showing the expression of HSPC marker *runx1* in the DA in *pkd2*[−/−] or *ift88*[−/−] with a sub-effective dose of *pkd2* MO (0.5 ng per embryo, *pkd2* MO_low) or *ift88* MO (0.8 ng per embryo, *ift88* MO_low). The red arrowheads indicate the expression of HSPC marker *runx1*. Scale bars, 100 μm

(Supplementary Fig. 1B and 1C). Together, the results suggest that cilia deficiency did not grossly affect the development of blood vessels and neighboring niche cells.

To explore the possibility that the HSPC defects observed in cilia-impaired embryos might be due to abnormal cell proliferation or apoptosis, we performed BrdU staining and TUNEL assay in Tg(*fli1a*:EGFP) embryos at 26 and 36 hpf (Supplementary Fig. 10A–D), respectively. The results showed that no obvious alterations of cell proliferation and apoptosis were present in blood vessels in *fsd1* morphants. As previously reported, injection of antisense MOs often induces a non-specific *p53*-dependent apoptosis[41]. To address this issue, *fsd1* aMO was injected into *p53* mutant embryos at one-cell stage. WISH results demonstrated that expression of *runx1* was still decreased in *fsd1* morphants (Supplementary Fig. 10E), indicating that the HE and HSPC defects were specifically due to *fsd1* deficiency. Altogether, these observations support the role of cilia in HE specification, independent of artery formation, niche cells, and EC proliferation or apoptosis.

**Cilia regulate HSPC emergence through Notch signaling.** To investigate the underlying molecular basis of how cilia impact HSPC emergence, we performed RNA-sequencing (RNA-Seq)

with the dissected trunk region from control and *fsd1* morphants at 26 hpf. Gene ontology and volcano plot analysis showed that among many developmental pathways, Notch signaling was mostly downregulated in *fsd1* morphants, compared to controls (Fig. 6a, b). qPCR analysis further confirmed dramatic reduction of a group of Notch target genes in *fsd1* mutants at 26 hpf, when primary cilia defects occurred (Fig. 6c and Supplementary Fig. 2A–C). Previously, it has been reported that Notch-processing enzymes and receptors co-localized with cilia and cilia stimulated Notch signaling in vertebrates[30,31]. Confocal imaging of the Notch activity reporter line Tg(*tp1*: mCherry/βact:Arl13b–GFP) showed that *tp1*[+] ECs in the ventral wall of DA are ciliated (Supplementary Fig. 11A). Next, we asked whether Notch signaling is involved in the cilia regulation of HE specification. First, the reduced *hey2* expression in DA of AGM region (Supplementary Fig. 11B) was observed. Western blotting analysis showed that the Notch intracellular domain (NICD) in *fsd1*[−/−] was decreased (Fig. 6d; Supplementary Fig. 11C). Second, to determine whether the Notch signaling in HE cells was altered, we examined the expression of notch receptor *notch1a* in *kdrl*[+] *runx1*[+] cells. The qPCR result showed that *notch1a* expression was decreased in HE cells (Fig. 6e), indicating that HE cells were sensitive to Notch

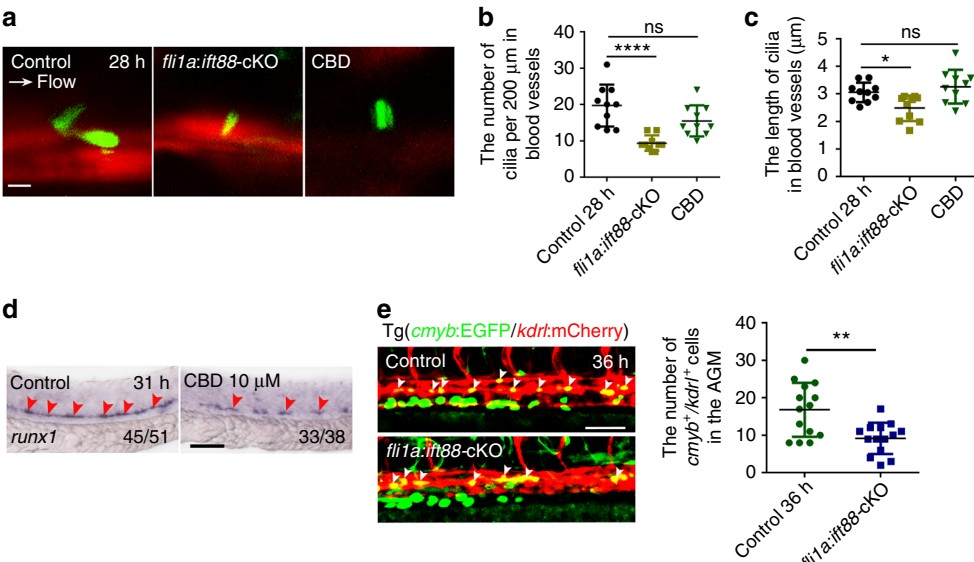

**Fig. 4** Blocking formation or function of primary cilia impairs hematopoietic stem and progenitor cell (HSPC) development. **a** Confocal imaging of cilia in the aorta-gonad-mesonephros (AGM) region in *kdrl*:mCherry/*βact*:Arl13b–GFP double-transgenic line with *fli1a*:*ift88*-cKO-functional injection, or ciliobrevin D (CBD) treatment at 28 hpf. White arrow denotes the blood flow direction. Scale bar, 5 μm. **b**, **c** The quantification of primary cilia number and length in the blood vessels of AGM in **a**. Data in **b**, **c** represent the analysis results of one-way ANOVA–Dunnett test. Error bars, mean ± s.d., *n* = 10 embryos. ns non-significant, *P < 0.05, ****P < 0.0001. **d** WISH result of *runx1* (red arrowheads) in control and CBD-treated embryos at 31 hpf. The red arrowheads indicate the expression of HSPC marker *runx1*. Scale bar, 100 μm. **e** The *kdrl*:mCherry⁺/*cmyb*:EGFP⁺ HE cells (white arrowheads) in the AGM region in *fli1a*:*ift88*-cKO-injected embryos (left panel) with quantification (right panel) at 36 hpf. Cmlc2:EGFP-negative embryos were used as a negative control (control). Scale bar, 50 μm. Error bars, mean ± s.d., *n* = 14 embryos. **P < 0.01, Student's *t*-test

signaling. Furthermore, the number of *fli1a*⁺*tp1*⁺ double positive cells was quantified during the HSPC production process in Tg(*tp1*:mCherry/*fli1a*:EGFP) embryos, and the results showed a reduced number of Notch-active ECs upon *fsd1* KD, compared to controls (Supplementary Fig. 11D). Consistently, the Notch activity was also reduced in the ECs of the other three cilia-impaired embryos (Fig. 6f, g; Supplementary Fig. 11E). Then, to determine whether cilia and Notch signaling act in the same pathway in hematopoiesis, the cilia-impaired embryos were treated with DBZ, a chemical Notch inhibitor. The expression of HE marker *runx1* showed obvious reduction in the DBZ treated embryos, but no more severe phenotypes observed in the cilia-impaired embryos plus DBZ treatment (Supplementary Fig. 11F and 11G), suggesting that cilia and Notch regulate HSPC development likely through the same pathway. To further investigate whether cilia transduce Notch directly, we applied *FSD1* KD approach using short interfering RNA oligonucleotides (siRNA) in human hTERT-RPE-1 (RPE-1) cells, and then performed immunofluorescence and western blotting (Fig. 6h, i). RPE-1 cells were transfected with GFP-NICD plasmid and si*FSD1*, respectively. The western blotting analysis showed that the protein levels of transduced GFP-NICD plasmid in control RNA (siCtrl) and si*FSD1* transfected cells were similar (Fig. 6i). Immunofluorescence staining and statistical results showed that the percentage of GFP-positive cells with nuclear NICD, as well as the GFP-positive RPE-1 cells with primary cilia was reduced in si*FSD1* group, compared to siCtrl group (Fig. 6j, k). Finally, over-expression of NICD specifically either in ECs by injecting the *fli1a*:NICD-EGFP vector (NICD-EGFP was driven by a *fli1a* promoter) or globally by injecting the *hs*:NICD-EGFP vector (NICD-EGFP was driven upon heat shock treatment) together with Tol2 mRNA, respectively[26], rescued the reduced *runx1* expression in cilia-impaired embryos (Fig. 7a–e; Supplementary Fig. 11F and 11G). Together, these data showed that Notch

signaling acts downstream of cilia to regulate HSPC development (Supplementary Fig. 11H).

## Discussion

Cilia have been extensively studied in a variety of biological processes including cell cycle, mechanosensation, chemosensation, and left-right asymmetry, during embryogenesis and in adulthood. However, its role in developmental hematopoiesis is poorly understood. In this study, we have identified that endothelial primary cilia have essential roles in HE specification during HSPC development in zebrafish embryos. Loss of cilia genes decreased the number and length of primary cilia in blood vessels of the AGM region, which led to a subsequent reduction of Notch signaling. Overexpression of NICD restored the HSPC defects in cilia-impaired embryos. Therefore, loss of primary cilia hindered Notch signaling transduction, which subsequently caused HSPC defects.

Accumulating studies demonstrated that primary cilia are involved in defining stem cell phenotypes by maintaining their 'stemness'. For example, primary cilia have been shown to influence the recruitment of mesenchymal stem cells (MSCs) through the regulation of transforming growth factor-β (TGFβ) signaling[42], and can also affect the process of stem cells differentiating into cardiomyocytes[43]. Furthermore, Shh signaling regulates adult neuron stem cell formation through primary cilia[11]. Although previous studies suggested the role of primary cilia in different types of adult stem cells in development, the exact regulatory process and the underlying mechanisms remain incomplete[8,42,44]. Here, our work initially defined a role of primary cilia, in HSPC emergence and the detailed analysis indicated that the bridge between cilia and HSPC development is mediated mainly through Notch signaling in the ECs of DA in zebrafish embryos.

Interestingly, HE specification defects in *fsd1* morphants and mutants are independent of artery formation and vessel integrity.

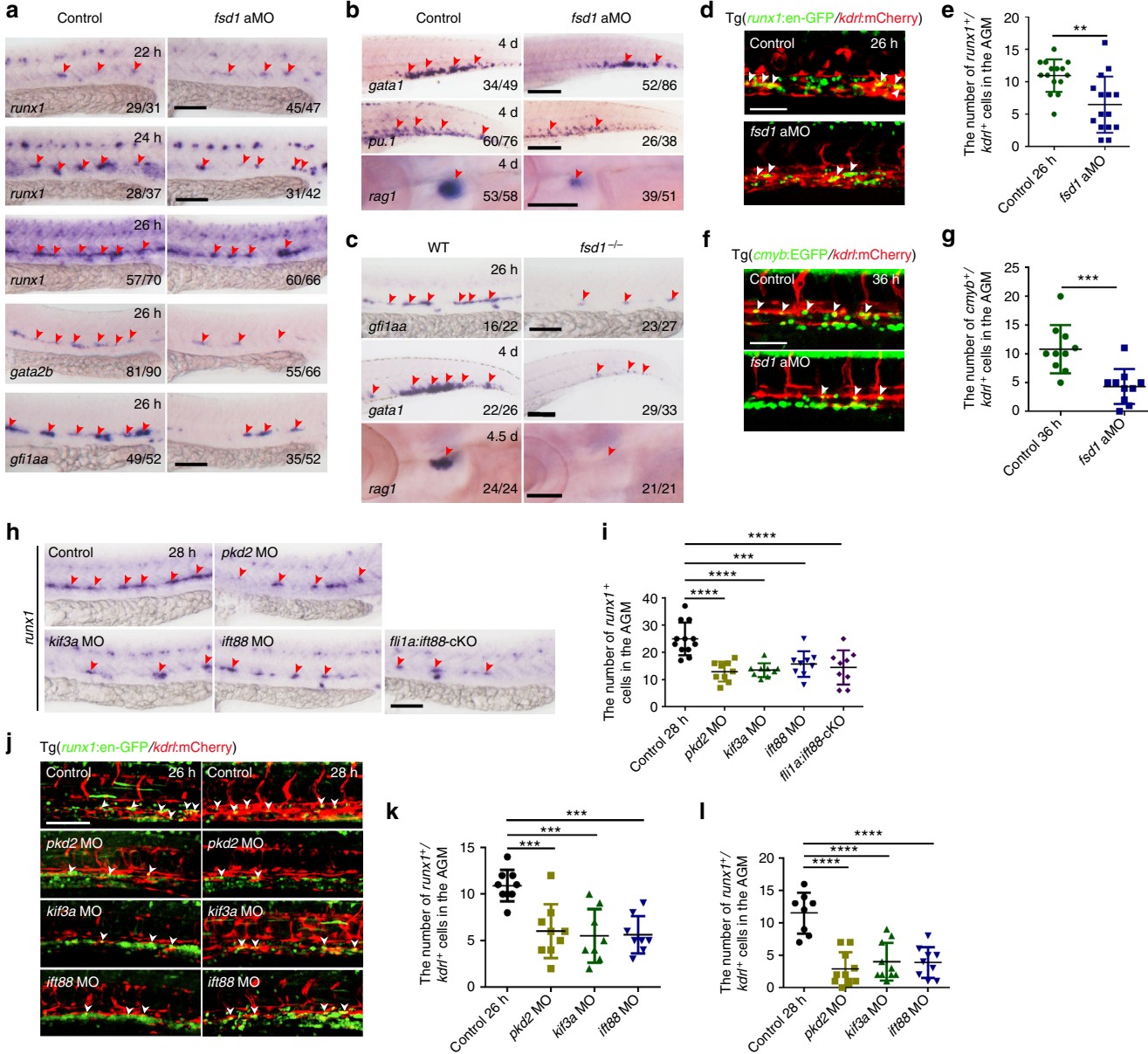

**Fig. 5** Cilia are required for hemogenic endothelium (HE) specification. **a** Whole-mount in situ hybridization (WISH) analysis showing the expression of HE markers, *runx1*, *gata2b*, and *gfi1aa*, in the aorta-gonad-mesonephros (AGM) region at 22, 24, and 26 hpf. Red arrowheads denote the expression of HE markers. **b** Expression of erythroid marker (*gata1*), myeloid marker (*pu.1*) in the caudal hematopoietic tissue (CHT) and T cell marker (*rag1*) in the thymus at 4 dpf. Red arrowheads mark the corresponding hematopoietic cells. **c** The expression of HE marker *gfi1aa*, erythroid marker *gata1*, and T cell marker *rag1* in *fsd1*−/−. Red arrowheads mark the corresponding hematopoietic cells. **d**, **e** The imaging of *kdrl*:mCherry+/*runx1*:en-GFP+ HE cells (white arrowheads) in the AGM region in control and *fsd1* morphants with quantification (**e**) at 26 hpf. Error bars, mean ± s.d., n = 15 embryos. **P < 0.01, Student's t-test. **f** Confocal imaging of HE cells (*kdrl*:mCherry+/*cmyb*:EGFP+) in control and *fsd1* morphants at 36 hpf. White arrowheads denote HE cells. **g** The quantification of *kdrl*:mCherry+/*cmyb*:EGFP+ HE cells in control and *fsd1* morphants at 36 hpf. Error bars, mean ± s.d., n = 10 embryos. ***P < 0.001, Student's t-test. **h**, **i** Representative images of *runx1* expression (red arrowheads) in control and cilia-impaired embryos with quantification (**i**) at 28 hpf. Red arrowheads denote *runx1*+ cells in the AGM region. **j–l** High-resolution imaging of HE cells (*kdrl*:mCherry+/*runx1*:en-GFP+) in control and cilia-impaired embryos with quantification (**k**, **l**). White arrowheads indicate *kdrl*:mCherry+/*runx1*:en-GFP+ cells in the AGM region. Data represent the analysis results of one-way ANOVA–Dunnett test. Error bars, mean ± s.d., n = 12, 9, 9, 9, 9 embryos (**i**). n = 9, 9, 8, 8 embryos (**k**). n = 8, 10, 10, 9 embryos (**l**). ***P < 0.001, ****P < 0.0001. Scale bars, 100 μm

We demonstrated that HE markers, including *runx1*, *gata2b*, and *gfi1aa*, were markedly downregulated, whereas the expression of artery markers *dll4* and *ephrinB2a*, and the EC number of artery and veins remained unaltered. This phenomenon is consistent with recent reports, in which artery identity can be uncoupled from HE specification and HSPC emergence[18,22,45,46]. We speculate that Notch signaling is differentially and dynamically required in these different processes, such as *dll4* and *ephrinB2a* in artery formation and *jagged1a* in HSPC emergence[18].

Once emerged, HSPCs can give rise to different blood cell lineages, and these processes are tightly controlled by a group of master transcription factors and signaling pathways. Notably, the coordination of cell intrinsic and extracellular signals has great impact on the HSPC formation and developmental hierarchy

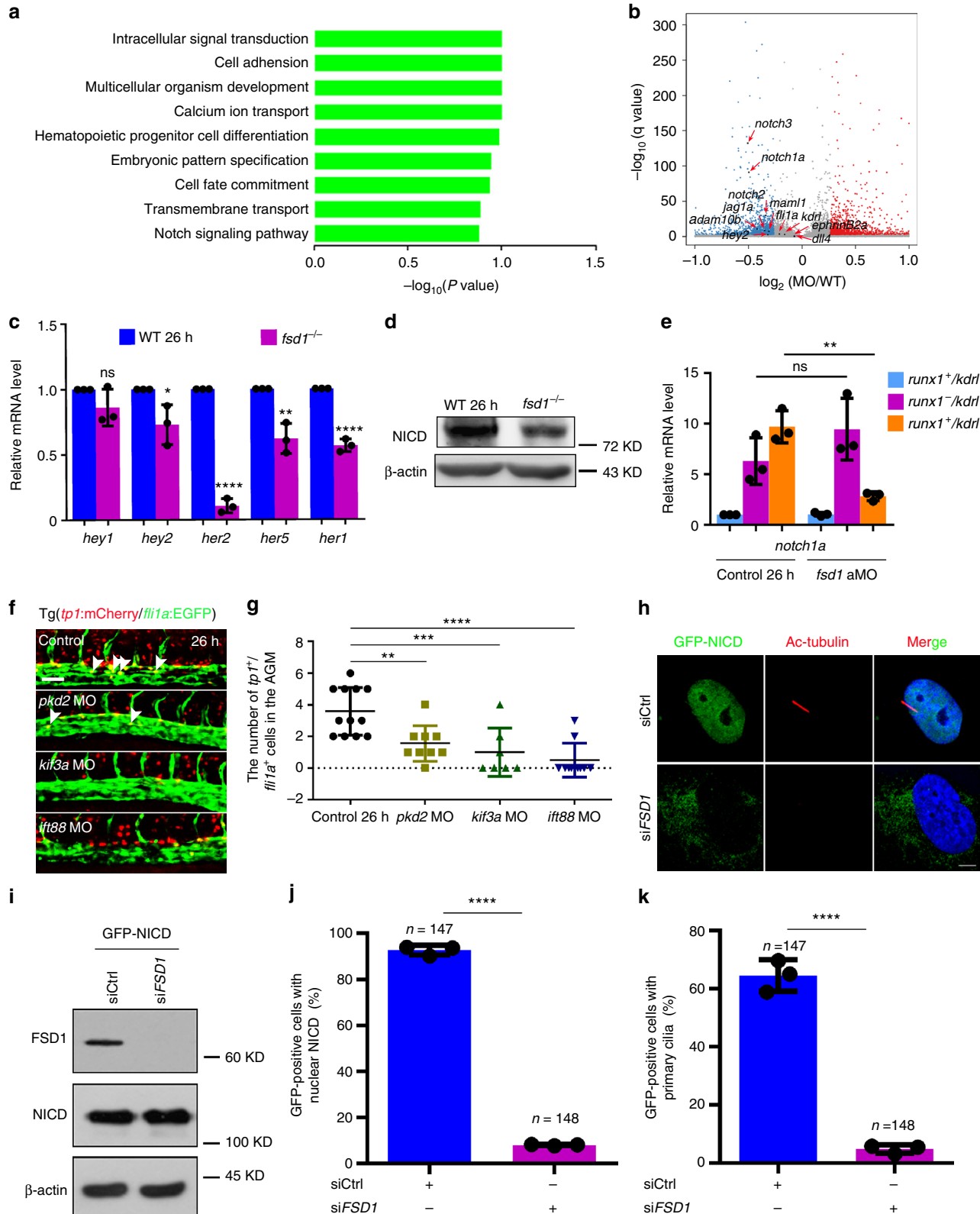

generation[47]. The well-known communication mode between cells and their niches is mediated by the signaling ligand/receptor pairs. Here, we found that the hairy-like organelle, primary cilium, can also transduce developmental signals into the cells from their niche. In brief, primary cilia act as communication hubs to transduce Notch signaling to emerging HE cells and HSPCs. Although the cilia-transduced signals exert a significant

influence on the cell cycle, organ formation, proliferation, differentiation, and embryonic development[1,8,48,49], the role of cilia in developmental hematopoiesis has never been studied before. Hence, this is the report to show how cilia modulate HSPC development in vertebrates.

The upstream regulatory factors of primary cilia in hematopoiesis are still unknown. Previous studies have shown that

**Fig. 6** Notch signaling is downregulated in hemogenic endothelium (HE) cells in primary cilia-impaired embryos. **a** The gene ontology analysis for the downregulated genes in *fsd1* morphants. **b** Volcano plot showing the dysregulated genes between wild-type embryos and *fsd1* morphants at 26 hpf. Endothelial cell markers *kdrl* and *fli1a*, arterial markers including *ephrinB2a*, *dll4*, and *hey2*, and Notch signaling components including *notch1a*, *notch2*, *notch3*, *maml1*, *jag1a*, and *adam10b* are highlighted in black. Red arrows denote the gene location site. **c** qPCR result of Notch target genes in wild-type embryos and *fsd1* mutants at 26 hpf. Error bars, mean ± s.d., * *P* < 0.05, ** *P* < 0.01, **** *P* < 0.0001, ns non-significant, Student's *t*-test. *n* = 3 biological replicates. **d** Western blotting showing the protein level of NICD in wild-type embryos and *fsd1*−/− at 26 hpf. **e** Expression of *notch1a* in *kdrl*+ *runx1*+ cells in the AGM in control and *fsd1* morphants at 26 hpf. Error bars, mean ± s.d., ** *P* < 0.01, ns non-significant, Student's *t*-test. *n* = 3 biological replicates. **f**, **g** Confocal imaging of Tg (*tp1*:mCherry/*fli1a*:EGFP) embryos in the AGM region in control and cilia-impaired embryos at 26 hpf. *tp1*:mCherry+/*fli1a*:EGFP+ double positive cells (white arrowheads) were quantified in **g**. Scale bar, 50 μm. Data represent the analysis results of one-way ANOVA–Dunnett test. Error bars, mean ± s.d., *n* = 12, 9, 7, 10 embryos, ** *P* < 0.01, *** *P* < 0.001, **** *P* < 0.0001. **h** Immunofluorescence staining of RPE-1 cells with acetylated α-tubulin (Ac-tubulin, cilia marker) antibodies and Hoechst (DNA marker) in ciliated and non-ciliated cells treated with siCtrl and si*FSD1* respectively. RPE-1 cells were transfected with GFP-NICD plasmid and indicated siRNA. The nuclei were stained with Hoechst (blue) and the cilium was marked by Ac-tubulin (red). The GFP indicated the NICD expression. Scale bar, 5 μm. **i** Western blotting showing the protein levels of NICD, β-actin and FSD1 in GFP-NICD plasmid transfected in siCtrl or si*FSD1* RPE-1 cells. **j**, **k** Quantification of GFP-positive cells with nuclear NICD (**j**) or with primary cilia (**k**) in **h**. Data are presented with three independent experiments. Error bars, mean ± s.d., **** *P* < 0.0001. Student's *t*-test. *n* number of cells

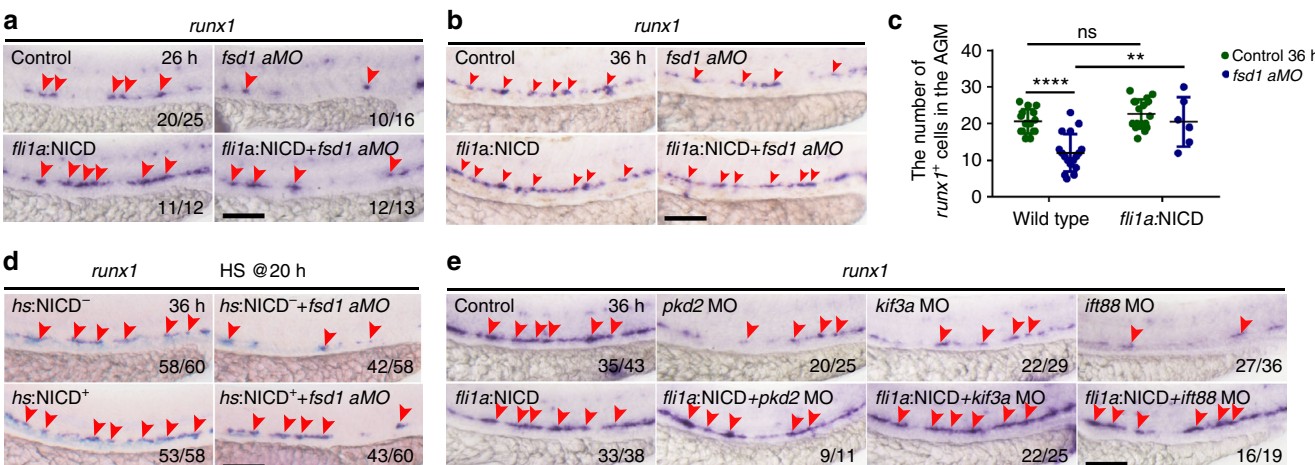

**Fig. 7** Notch signaling acts downstream of primary cilia in regulating hematopoietic stem and progenitor cell (HSPC) specification. **a**, **b** Whole-mount in situ hybridization (WISH) result of *runx1* expression in control embryos, *fsd1* morphants, *fli1a*:NICD-EGFP plasmid-injected embryos, and *fsd1* morphants injected with *fli1a*:NICD-EGFP plasmid at 26 hpf and 36 hpf. Red arrowheads denote the *runx1*+ cells in the AGM region. **c** Quantification of *runx1*+ cells in the aorta-gonad-mesonephros (AGM) region in **b**. Data represent the analysis results of two-way ANOVA–Tukey test for multiple comparisons. Error bars, mean ± s. d., *n* = 15, 17, 17, 6 embryos. ** *P* < 0.01, **** *P* < 0.0001. **d** WISH result of *runx1* expression (red arrowheads) in control embryos, *fsd1* morphants, *hs*:NICD-EGFP plasmid injected embryos, and *fsd1* morphants injected with *hs*:NICD-EGFP at 36 hpf. These embryos were heat shocked for 30 min at 42 °C from 20 hpf. **e** Expression of *runx1* (red arrowheads) in control, morphants, and *fli1a*:NICD-EGFP injected morphants at 36 hpf. Scale bars, 100 μm

endothelial primary cilia can mediate blood flow in chicken heart[50], mouse heart[51], and zebrafish blood vessels[12]. It has been reported that blood flow is required for embryonic HSPC development[52–54]. Furthermore, we and others recently showed that the Notch target gene (*ephrinB2a*) was downregulated in blood flow-deficient embryos[52,53]. Therefore, it is reasonable to hypothesize that blood flow may lie upstream of primary cilia. Further investigation is required to validate this notion.

In summary, we demonstrated that the hairy-like organelle, primary cilium, is required for HE specification in HSPC development via regulation of Notch signaling. This finding further deepens our understanding on the role of cilia in HSPC development, and may provide insights into developing new strategies in generating functional HSPCs for clinical application.

## Methods

**Zebrafish strains**. Adult Zebrafish including AB strain, Tg(*cmlc2*:GFP), Tg(*runx1*: en-GFP), Tg(*gfi1*:GFP)[55] Tg(*fli1a*:EGFP) (provided by Steve Wilson), Tg(*kdrl*: mCherry), Tg(*cmyb*:EGFP), Tg(*tp1*:mCherry), Tg(*fli1a*:nGFP) (provided by Anming Meng), Tg(βact:Arl13b–GFP)[56], *ift88* mutant[38], *pkd2* mutant (provided by Chengtian Zhao), *p53* mutant[57] (provided by Anming Meng), and *fsd1* mutant were maintained in 28.5 °C system water in the Institute of Zoology, Chinese Academy of Sciences. *fsd1* mutant with a 20 bp insertion in the fourth exon of *fsd1* was generated by CRISPR/Cas9, and the primers for genotyping are listed in

Supplementary Table 1. The embryos were all natural spawning of adult zebrafish. This study was approved by the Ethical Review Committee in the Institute of Zoology, Chinese Academy of Sciences, Beijing, China.

**Morpholinos**. MOs in this study were ordered from Gene tools and were injected into 1–2 cell-stage embryos at the boundary of yolk and cell. The sequences of MOs used in this study are listed in Supplementary Table 2.

**Plasmid and mRNA synthesis**. We constructed a *fli1a*:*fsd1*-EGFP plasmid, which expresses the capped fish *fsd1* mRNA in ECs under the control of *fli1a* promoter within an EGFP reporter. The capped full-length CDS of zebrafish *fsd1* was cloned into a pDONER221 plasmid through using Gateway BP reaction (Gateway BP Clonase II Enzyme mix, Invitrogen), and then sub-cloned into pDestTol2pA2 with a *fli1a* promoter and an EGFP reporter by Gateway LR reaction (LR Clonase II Plus enzyme; Invitrogen). The human full-length *fsd1* mRNA (hmRNA) was synthesized from a pCS2 plasmid using mMessage mMachine SP6 kit (Ambion). The *fli1a*:*fsd1*-EGFP plasmid (30 pg) and *fsd1* hmRNA (25 pg) were injected into one-cell stage embryos alone or in combination with *fsd1* MOs. The GFP-NICD plasmid was generated by cloning the sequence encoding mouse NICD1, fused with GFP sequence in the N-terminal region, into the pCS2 vector.

The *fli1a*:*ift88*-cKO construct was provided by Massimo M. Santoro (University of Turin, Turin 10126, Italy). In this construct, the *fli1a* promoter was cloned into the pDestTol2pA2-U6:gRNA (guided RNA) (Addgene #63157) by the gateway system. Then the guide RNA-seq of *ift88* was inserted into the above plasmid to generate the final *fli1a*:*ift88*-cKO vector[40]. We validated the efficiency of *fli1a*:*ift88*-cKO using T7E1 assay in sorted *fli1a*+ cells[40]. We extracted the genomic DNA and amplified the target region by PCR. The PCR products were incubated at 37 °C for

60 min with T7E1 enzyme and examined by 2.5% agarose gel. The primers are listed in Supplementary Table 3.

**Chemical treatment**. Live zebrafish embryos were incubated with DBZ[27] (8 μM, Sigma), CBD (Ciliobrevin D, 10 μM, Merck) from 10 hpf to the examined time point to ensure pharmacologic effect.

**Whole-mount in situ hybridization**. Whole-mount in situ hybridization (WISH) was performed through standard procedure[58]. The following probes, including *runx1, cmyb, gata2b, gfi1aa, gata1, pu.1, scl, rag1, hey2, ephrinB2a, dll4, msr, flt4, myod, kdrl*, and *pax2.1* were used in this study[18,27,59]. Nikon SMZ 1500 microscope was used to collect figures.

**Western blotting**. The dissected trunk regions of zebrafish embryos were collected for extracting protein. Western blotting was performed[60] with 12% sodium dodecyl sulfate–polyacrylamide gel electrophoresis. The proteins were then transferred into a nitrocellulose membrane and then membranes were blocked by nonfat milk. After blocking, the nitrocellulose membranes were then incubated using anti-Fsd1 antibody (Ab) (1:200, monoclonal mouse or polyclonal rabbit anti-FSD1 antibody was generated using a glutathione-S-transferase fusion protein containing the full-length FSD1, which were expressed in E. coli and then were purified to homogeneity as the antigen. The monoclonal mouse Ab was used in human RPE-1 cells, while the polyclonal rabbit Ab was used in zebrafish experiments), anti-Runx1 Ab (1:200, AS-55593, Ana Spec), anti-NICD Ab (1:400, ab83232, Abcam, used in zebrafish), anti-NICD Ab (1:1000, ab8925, Abcam, used in human cells), mouse anti-β-actin (1:1000, sc47778, Santa Cruz Biotechnology, used in human cells), and anti-β-actin Ab (1:2000, 4967, Cell Signaling). The figures for western blotting were collected by Imagequant LAS 4000 (The uncropped scans of blots are listed in the Supplementary Figure 12).

**Cell sorting and quantitative RT-PCR**. ECs, HE cells, and HSPCs from the dissected trunk region of *kdrl*:mCherry/*runx1*:en-GFP transgenic embryos at 26 hpf were sorted by fluorescence-activated cell sorter[27]. Total RNAs from the sorted cells of wild-type embryos and *fsd1* morphants were extracted using QIAGEN RNeasy Micro kit and then were transcribed. Total RNA from the dissected trunk regions was extracted with TRIzol reagent (Ambion) and reversely transcribed. The cDNA was diluted and used for templates. The qPCR was performed[52], and the primers used for qPCR are shown in Supplementary Table 4.

**RNA-seq**. Dissected trunk region of wild-type embryos and *fsd1* morphant at 26 hpf were collected for extracting RNAs. The RNAs were then reversely transcribed and amplified cDNA were sequenced on a BGISEQ-500 in Beijing Genomic Institution.

**Confocal microscopy**. Embryos were mounted on dishes with 1% low-melting agarose. Live embryos were anesthetized before mounting. Confocal images were captured by a Nikon A1 confocal laser microscope. The analysis of the images was carried out by Nikon confocal software, Image J and Imaris.

**BrdU assay and TUNEL staining**. BrdU (10 mM) was injected into the yolk of *fli1a*:EGFP transgenic embryos. The embryos were fixed by 4% PFA 2 h later and then treated with standard procedure[60]. Briefly, the embryos were washed three times with 1 × PBST 5 min each and then treated with Proteinase K (10 μg/mL) and 2 N HCl. After blocking with 1% BSA for 1 h, the embryos were incubated with anti-BrdU antibody (1:800, 5-Bromo-2′-deoxy-uridine Labeling and Detection Kit, Roche). Before imaging with the confocal microscopy, the embryos were washed and incubated with anti-mouse-Ig-fluorescein (1:500, 5-Bromo-2′-deoxy-uridine Labeling and Detection Kit, Roche).

TUNEL staining was performed[52]. In brief, embryos were fixed by 4% PFA for at least 12 h at 4 °C and then were dehydrated with methanol for at least 4 h at −20 °C. Then the dehydrated embryos were washed three times with 1 × PBST 5 min each and following Proteinase K treatment. The embryos were refixed by 4% PFA at room temperature for 20 min and washed three times with 1 × PBST 5 min each. The embryos were then incubated in 50 μL mixture (In Situ Cell Death Detection Kit, Fluorescein; Roche) overnight at 4 °C. Finally, the embryos were washed three times with 1 × PBST 5 min each and then were captured by confocal microscope.

**Transmission electron microscope**. Transmission electron microscope (TEM) was performed in control and *fsd1* morphants at 28 hpf. In brief, zebrafish embryos were fixed by 2.5% glutaraldehyde in 0.1 M phosphate buffer solution (PBS, pH 7.2) and were then postfixed by 1% osmium tetroxide for 2 h. After washing by ethanol with different gradients, embryos were dehydrated with different gradients of acetone and epoxy resin. Then, embryos embedded in resin were ultrathin sectioned (60 nm) and the sections were mounted on copper slot grids for further JEM1400 electron microscope imaging.

**Cell culture**. Human hTERT-RPE-1 (RPE-1) were cultured in DMEM/F-12 (1:1) medium, supplemented by 10% FBS, 1% penicillin/streptomycin B and 0.01 mg/mL hygromycin. RPE-1 cells were then serum starved for 48 h in Opti-MEM reduced serum media (Life Technologies) for cilia generation.

**Immunofluorescence**. RPE-1 cells were transfected with GFP-NICD plasmid with Lipofectamine LTX Reagent (Life Technologies) according to the manufacturer's instruction in RPE-1 cells. 6 h post transfection of GFP-NICD plasmid in RPE-1 cells, the medium was changed. RPE-1 cells were transfected with control (siCtrl) or FSD1 siRNA (siFSD1) at 24 h post plasmid transfection. 24 h post siRNA transfection, the RPE-1 cells were starved for 48 h. The cells were then fixed with 4% paraformaldehyde for 10 min at room temperature. After blocking, the RPE-1 cells were incubated with mouse anti-Ac-tubulin (1:400, T6793, Sigma) Ab. The DNA was stained with Hoechst 33342 (1:500, H3570, Invitrogen). The RPE-1 cells were then incubated with second antibodies. The images were captured by Zeiss LSM 880 and were analyzed with Volocity 6.0. The sequences of siRNAs are shown in Supplementary Table 5.

**Statistical analysis**. Student *t*-test, one-way ANOVA and two-way ANOVA were used in the statistical analysis. The statistical results were analyzed by GraphPad Prism 6.01 software. Data were represented as mean ± SD, *$P < 0.05$, **$P < 0.01$, ***$P < 0.001$, ****$P < 0.0001$.

**Reporting Summary**. Further information on experimental design is available in the Nature Research Reporting Summary linked to this article.

## Data availability

The RNA-seq data presented in the paper were deposited at NCBI SRA database under accession number SRP140462. Source data is provided as a Source Data file. All the relevant data supported this study are available from the corresponding author upon request.

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

## Acknowledgements

We thank Steve Wilson, Massimo M. Santoro, and Anming Meng for fish lines and reagents, and the Liu lab members for providing critical feedback on the manuscript. This work was supported by grants from the Ministry of Science and Technology of China (2016YFA0100500), the Strategic Priority Research Program of the Chinese Academy of Sciences, China (XDA16010207), and the National Natural Science Foundation of China (31830061, 31425016, and 81530004).

## Author contributions

Z.L. performed experiments with help from L.W., Y.X. and D.M.; H.T. and H.L. provided Fsd1 antibody and performed experiments in cell culture; Y.K. and C.Z. provided reagents; Z.L. and F.L. conceived the project, analyzed the data, and wrote the paper. All authors read and approved the final manuscript.

## Additional information

**Competing interests:** The authors declare no competing interests.

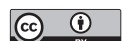

