## [Peer Review File · Nature Communications]

Reviewers' comments:

Reviewer #1 (Remarks to the Author):

In this manuscript, Liu and colleagues assess the role of primary cilia in the process of hematopoietic stem cell formation in zebrafish. The authors propose that primary cilia do so by modulating notch signaling. The fact that cilia modulates HSC generation is potentially interesting, nevertheless the results are not up to the standards in the zebrafish community.

Today, many researchers would not trust a MO for studies of vascular or HSC development; if a mutant is available, the authors should definitely use it (especially since this allows one to do experiments in a blind way). Here is an excerpt of the most recent guidelines to the use of morpholino in zebrafish (Stainier et al., PLoS Genetics, 2017) 'Finally, a word of caution that previous publication of MOs is not a guarantee of their fidelity, particularly if a new phenotype is being described'. In addition, the fact that cilia can modulate notch signaling has already been proposed (See Samsa et al., Development, 2015). Thus, in that aspect the link between notch and cilia is not completely new. Furthermore, it is now well accepted that the endothelium is ciliated at times of HSC generation in fish. Thus overall, it is unclear if the present manuscript provides a clear new concept in the field. Thus, the manuscript needs major revisions to match the current standards and provide convincing evidences of their claims.

Specific comments

1. the authors must assess the phenotype using the known mutants for *pkd2*, *ift88*. It needs to be demonstrated that the MO phenotype is recapitulated in the mutants. Along these lines, it is well known that *ift88* has cilia independent function in cells and zebrafish (see Delavale et al. 2011, 2017 or Vertii et al., 2015). The authors should analyse other cilia mutants to confirm the observations made with *ift88* MO such as *kif3a* mutants (which would also validate the *kif3a* MO data).
2. Similarly, in depth *fsd1* mutants analysis should be provided. This is extremely important as the MO has been used to perform the mRNA seq analysis. The mutant is mentioned in the methods but the phenotype of the mutants is described extremely superficially in sup results. The authors should show the complete mutants analysis (same as the MO) in the main figures not the MO.
3. The most common way cilia function is to modulate *shh* signaling. Surprisingly, the authors completely occult the fact that Hh has a major role in HSC emergence (see for example Wilkinson et al., Dev Cell 2009). It is necessary that the authors clarify if Hh is involved here and clarify if Notch acts downstream of Hh in that context.
4. The authors previously showed that *klf2a* is necessary for HSC generation. Even though *klf2a* function has recently been challenged in HSC generation (see Novodvorsky P et al., PLoS One, 2015), it is surprising that the authors do not analyse *klf2a* expression in this context.
5. The authors (and others) have shown that the EdCs are ciliated, but they have not shown that HSCs are ciliated. This needs to be clarified experimentally by performing detailed analysis of cilia on HSCs.
6. A related concerns, the authors mentioned that notch signaling act downstream of cilia, implying that cilia function cell-autonomously to regulate notch signaling. However, this is not assessed experimentally. This point is key for the conclusions and the authors should assess if cilia function cell autonomously or non-cell autonomously.

7. In the in situ data, the sample number is low even though the authors use MO which usually allows to obtain lots of embryos for the analysis. In that aspect, one expect the phenotype to fluctuate from embryo to embryo but the authors never mention any classification of the phenotypes and their variability? For example, in Figure 3A, which signal is pointed out by the red arrowhead? It is very unclear.

8. The authors performed injection the *fli1a:ift88-cKO* DNA. This transient knock-down approach is not completely appropriate to describe the *fli1a:ift88-cKO* in the Figure 4. Rescuing the *ift88* mutants with *fli:ift88* is a much better way to assess the tissue specificity of *ift88* function. In addition, considering *ift88* has cilia independent functions, *kif3a* or any other tissue specific rescue of the cilia mutants would be required.

Reviewer #2 (Remarks to the Author):

In this study, Liu et al show that embryonic endothelial cells have cilia which is important for the formation of hematopoietic progenitors in the zebrafish embryo. By using morpholinos and crispr technologies they disrupt different genes involved in cilia formation and report a hematopoietic phenotype. They further study the effect on hemogenic endothelium (HE) cells by using transgenic reporters (*runx1*, *kdr* or *c-myb,kdr*). They report a specific requirement of cilia in the formation of HE and in Notch activation.

Although the findings are interesting and novel, this work has several fundamental problems that would require a big amount of experiments to resolve and the current conclusion may not stand. The main concern is that the genetic mutants they analyze in most of the experiments are general mutants that may affect other cell types and other decisions. The phenotype on *Runx1* cells that the authors observe after morpholino treatment is only happening after certain stage (26hpf) and while many determinations are performed at 30 to 36 hpf, many vascular controls and other tissue integrity are performed at 26-28 hpf.

The specific comments for each figure are:

Figure 1

What is *Arl13b*? Is that a single cilia per cell what they show in the image and estimate from quantification? Could they give some information on that?

The authors performed the first analysis with a *Runx1* reporter, are all endothelial cells that they find with cilia negative for *Runx1*? That does not agree with the next figures, could they show *runx1+* cells with cilia?

Figure 2

What is the reference for the 'newly identified centrosomal gene *fsd1*' . Could they show staining for *fsd1* and the other cilia genes used?

As previously said, the phenotype of morpholinos is very broad, in Figure 2B the endothelial layer is disrupted the defects could come from cilia or from something else.

Figure 3

Figure 3supl: the population *kdr+runx+* has a strange expression pattern for *flk1* (negative?) while expressing a lot of *c-myb*. Similarly in *kdr-runx+* there seem to have very little *c-myb* expression. How were these populations isolated? How many replicates were done? Were they independent biological replicates?

Surprisingly the authors claim that *kdr+runx+* contain cilia but cells in Figure 1 were *runx1* negative or at least they never show that cilia is on the HE cells.

The authors show that there are less runx1+ and cmyb+ cells in the morpholino treated embryos, however they should show how does the general vasculature looks like in these mutants at the same time of the analysis.

Figure 4

In these experiments they KO itf88 in fli1 cells and find that hematopoietic cells are decreased, although it is not too obvious from the image showed in figure 4E. Why do they use itf88 instead of fsd1? Is it assumed that all these genes are doing the same but they may not, so it would be more informative to use the same ones for all the studies. In figure5 they focus again in fsd1 again.

Figure 5

Authors check expression of EphrinB2 and Dll4 in the morphants and find that vasculature is normal. However they check at 24 hpf, also at 26hpf, but most of their analysis are made at 31-32-36 hpf. Is the vasculature and arterial/venous makers still normal? Is the skeletal muscle still normal? Some images (Fig2) don't look so normal.

Figure 6

They perform RNA-seq experiments by dissecting the trunk of the embryo of control and fsd1 morphants. This strategy has the problem that many cell types were sequenced and it is unlikely that it represents the endothelium or HE population.

Next they perform a WB with N11C and find that active Notch1 has almost disappeared, but the quality of this blot is not great and it is surprising that it is abolished. Such abolishment of Notch activity should result in the loss of the arterial program, which at least at 26 hours was not lost, maybe it is lost at this time.

By the tp1 reporter they also show reduction of Notch activity, but not a complete abolishment. These data together shows some contradictions that are difficult to reconcile.

It is true that they rescue the runx1 expression in the mutants with the overactivation of Notch, but this is not too surprising since Notch can induce runx1 in different types of endothelial cells in the fish, so it may just be overriding any previous defect.

In general the authors should be more consistent in the time of analysis that they use and should specifically knock out the genes in the endothelium to study the effect on HE. The transcriptome and Notch studies should also be more specific.

Reviewer #3 (Remarks to the Author):

COMMENTS FOR AUTHORS

In this manuscript, Liu et al present experimental studies implicating cilia, via a contribution to notch signalling, as important for hematopoietic stem cell specification in the hemogenic endothelium. I really enjoyed reading this manuscript. It is very well presented and the studies very comprehensively test the hypothesis. The experimental strategy generally presents multiple lines of evidence, often that is technically independent, to support each point.

MAJOR COMMENTS

1. The studies are comprehensive, but exclusively conducted in zebrafish. For the generic claim of the

title to be true, some evidence that the pathway is conserved is required i.e. that it applies in mammalian systems also. Alternatively, the title would be acceptable without further experiments if it restricted the claim to zebrafish.

2. A role for Notch signalling in HSC specification is well established, including in zebrafish experimental systems, and so the involvement of Notch signalling in HSC specification per se is not a discovery of this report. Nor is the localisation of Notch pathway components and cilia (as acknowledged in citations 24 and 25). So it is the linking of these two elements that is the novelty. The introduction and discussion would contextualise the contribution of this report better if it cited some of the Notch-HSC specification literature, particularly the zebrafish contributions (e.g. Berns et al, *Genes Dev* 2005; Bertrand et al, *Blood* 2010; Clements et al *Nature* 2011; and a review, Butko et al *Dev Biol* 2016). In fact, the present studies rescue HSC emergence in the absence of cilia by overexpressing Notch - that is, presuming that cilia presence is not itself restored by enforced Notch expression. Whether this is occurring or not is not experimentally addressed. If it is not, the hemogenic endothelial cilia may be the usual site of the Notch contribution, but are not the only possible site.

3. The *fli1a:ift88-cKO* construct is a good experiment, achieving tissue-specific CRISPR/cas9 tissue-specific knockdown, of which there are only a few examples to my knowledge in zebrafish.

3a. Insufficient methodological detail is provided for others to replicate this experiment (sFig5 is not enough). What form of cas9 was used? What was the sequence of the gRNA in the transgene? Which U6 promoter was used? When the genotyping primers for which the sequences are provided are used, what is the distinguishing outcome?

3b. The label sFig5C was initially confusing for me – it would have helped if it had been labelled to show that the genotype was *Tg(kdrl:mCherry/cmyb:EGFP) + (the construct in B)*. Although I realise that *flk1* (used in the legend) and *kdrl* (used in the figure) are the same (by referring to ZFIN ID: ZDB-GENE-000705-1), it is unnecessarily confusing, particularly to those not in the field, to use both interchangeably like this. Similarly, there is use of both terms in sFig3.

4. Following on from this, in sFig3A it was surprising that the level of *flk1*(=*kdrl*) expression was so low in the *runx1+kdrl+* (yellow) population. Why is this so?

OTHER COMMENTS

1. The Actin control for the Western blots in Fig 3 and SFig are, I think, the same image. While this is possible (both show control 36h and *fsd1* aMO lanes), this should be explicitly stated, particularly as the actin control in Fig 6E (which also has the same lanes) is not.

2. Descriptive statistics - It was often difficult to be sure of n-values, or what n-values meant, for the descriptive statistics.

2a. Fig 1C, D. No n values provided.

2b. For all other instances of graphs with y-axes labelled "The length of cilia in blood vessels (microm)", what is the n-value? For example, Fig 2F, it appears to be only n=4-6/group, which seems like too small a sample given the variation in cilia length shown for this variable in the adjacent Fig 2D – despite the statistical significance, the small n-values provide too much opportunity for biased selection.

3. Statistics- replication.

3a. While many observations are supported by multiple lines of evidence, and are hence independently corroborated, it is often unclear how many times an individual line of evidence was replicated. For example, in Fig 4e, there are 7 data points per group – were these all obtained from

one experiment in one day, which would be one experiment with $n=7$. Also, in this case, are the test data points unselected as the legend suggests by saying "fli1a:ift88-cKO-injected embryos", or were they selected for expression of the cmlc2-EGFP backbone marker in the transgene, which would make sense? The same query applies to Fig 4B and C, Fig 6I and in many supplementary figures.

3b. Fig 5J and K. These appear to be replicate experiments – was this intended? If so, the reason for presenting two experiments in this instance is not stated.

4. Statistics – other questions

a. Were the t-tests two-tailed or not (I see no reason why they should not have been 2-tailed).

b. In the instance where multiple pair-wise comparisons are being performed by multiple T-tests (as in sFig10B, but also elsewhere), an adjustment for multiple comparisons is required.

Editing.

1. Sometimes the y-axis label is provided as a title to a figure panel rather than adjacent the y-axis (Fig 1C,D; 4E (in contrast to B and C); Fig 5 D,F (in contrast to J, K); sFig 4C). I suggest consistent use as a y-axis label.

2. sFig1B. The axis label states "%" but the axis range is 0-1.5; either change the numbers to %s or use the term "proportion".

3. The English is excellent and requires very little editing.

Line 35 – delete "of"

Line 46 – add "a" following "led to"

Line 57 – last word "of" instead of "on"

Line 98 – space between "region in"

Line 105 – delete ", " after fsd1

Line 133 – Greek letter "mu" for "5um", not letter "U"

Line 170 – suggest "achieved" rather than "employed"

Response to reviewers' comments

We are very grateful to the handling editor and the three reviewers for their insightful comments on our manuscript. We have clarified the experimental details and performed a number of critical experiments to improve the quality of our manuscript, such as the detailed ciliogenesis analysis in HE, HSPC and ECs, tissue-specificity analysis, rescue experiments, vasculature assay and most importantly, using the other three types of cilia mutants (*ift88*^{-/-}, *pkd2*^{-/-} and *kif3a*^{-/-}) to confirm the HSPC phenotypes. The detailed point-by-point responses to reviewers' comments are shown below.

Major concerns

We hope you will find the referees' comments useful as you decide how to proceed. We would be happy to look at a substantially revised manuscript incorporating further experimental data or analysis that allows you to address these criticisms. It will be necessary to:

- Provide further support for the specificity of phenotypes described in this work to cilia and to the vasculature/HSCs (Reviewers #1 and #2)
- Describe the potential role of Hh in this context (Reviewer #1)
- Clarify experimental details (Reviewers #2 and #3)

Response to the editor's comments

Thanks for pointing out the critical issues for us to revise our manuscript. We have provided all the required experimental data to strengthen our conclusion of how cilia affecting HSPC development through Fsd1-Notch signaling. For details, please refer to our detailed responses to the reviewers' comments below.

Reviewer #1 (Remarks to the Author):

In this manuscript, Liu and colleagues assess the role of primary cilia in the process of hematopoietic stem cell formation in zebrafish. The authors propose that primary cilia do so by modulating notch signaling. The fact that cilia modulates HSC generation is potentially interesting, nevertheless the results are not up to the standards in the zebrafish community.

Today, many researchers would not trust a MO for studies of vascular or HSC development; if a mutant is available, the authors should definitely use it (especially since this allows one to do experiments in a blind way). Here is an excerpt of the most recent guidelines to the use of morpholino in zebrafish (Stainier et al., PLoS Genetics, 2017) 'Finally, a word of caution that previous publication of MOs is not a guarantee of their fidelity, particularly if a new phenotype is being described'. In addition, the fact that cilia can modulate notch signaling has already been proposed (See Samsa et al., Development, 2015). Thus, in that aspect the link between notch and cilia is not completely new. Furthermore, it is now well accepted that the endothelium is ciliated at times of HSC generation in fish. Thus overall, it is unclear if the present manuscript provides a clear new concept in the field. Thus, the manuscript needs major revisions to match the current

standards and provide convincing evidences of their claims.

Specific comments

1. the authors must assess the phenotype using the known mutants for **pkd2**, **ift88**. It needs to be demonstrated that the MO phenotype is recapitulated in the mutants. Along these lines, it is well known that *ift88* has cilia independent function in cells and zebrafish (see Delavale et al. 2011, 2017 or Vertii et al., 2015). The authors should analyse other cilia mutants to confirm the observations made with *ift88* MO such as *kif3a* mutants (which would also validate the *kif3a* MO data.)

Response 1: We thank this reviewer for pointing out this important issue. We have performed experiments to confirm the specificity of the observed HSPC defects in *ift88* or *pkd2*-deficient embryos and the results have been added into the revised manuscript (**Fig. 3F and Supplemental Fig. 5**).

From the WISH result, the expression of *runx1* was unchanged in *pkd2* mutants and *ift88* mutants¹ at 26hpf and 36hpf (**Supplemental Fig. 5E**), which is likely due to their maternal expression at early stages (pronephric duct cilia was unchanged in mutants, compared to wildtype siblings. **Supplemental Fig. 5D**). To avoid the maternal effects of *pkd2*^{2, 3}, we injected a sub-effective dose of *pkd2* MO into wildtype siblings and *pkd2* mutants, at which dose the MO could not cause HSPC defect in wildtype embryos (**Supplemental Fig. 5F**), then we observed *runx1* expression was reduced in *pkd2* mutants injected with low dose of MO at 36hpf (**Fig. 3F**). The similar decreased *runx1* expression was also seen in *ift88* mutants injected with a sub-effective dose of *ift88* MO (**Fig. 3F**). Therefore, the HSPC defects were also observed in *pkd2* and *ift88* mutants injected with low dose MOs, consistent with that in *pkd2* and *ift88* morphants. We also examined the T cell marker *rag1* at 4 dpf in *kif3a*^{-/-4} injected with low dose *kif3a* MO and in *ift88*^{-/-} with low dose *ift88* MO, and observed the decreased *rag1* expression (please see figures below). These results support that the HSPC differentiation is impaired in cilia-deficient embryos.

Collectively, we concluded that the cilia defective mutants largely recapitulate the phenotypes observed in cilia defective morphants.

2. Similarly, in depth *fsd1* mutants analysis should be provided. This is extremely important as the MO has been used to perform the mRNA seq analysis. The mutant is mentioned in the methods but the phenotype of the mutants is described extremely superficially in sup results. The authors should show the complete mutants analysis (same as the MO) in the main figures not the MO.

Response 2: Thank you for this thoughtful suggestion. We carefully examined the phenotypes of HE cells, HSPCs and HSPC derivatives in *fsd1* maternal-zygotic mutant (*fsd1*^{-/-}) (**Fig. 3C and Fig.5C**). Meanwhile, the grossly normal vasculature (*kdrl* marking pan-vessels, *dll4* and *ephrinB2a* marking artery and *flt4* marking vein, respectively) was also observed in *fsd1*^{-/-} (see the following figures). Mechanistically, we observed the decreased expression of Runx1 and NICD in *fsd1*^{-/-} (**Fig. 3D and Fig. 6D**), as well as decreased Notch target genes in *fsd1*^{-/-} by qPCR (**Fig. 6C**). In conclusion, the HE and HSPC defects, Notch signaling defect and normal vascular genesis in *fsd1*^{-/-} were consistent with that in *fsd1* morphants. We have added the mutant data in the main figures as you suggested (**Fig. 3C, 3D, 5C, 6D**).

3. The most common way cilia function is to modulate shh signaling. Surprisingly, the authors completely occult the fact that Hh has a major role in HSC emergence (see for example Wilkinson et al., Dev Cell 2009). It is necessary that the authors clarify if Hh is involved here and clarify if Notch acts downstream of Hh in that context.

Response 3: We thank this reviewer for pointing out this critical point. We examined the expression of genes in Hh signaling at 26hpf and 36hpf, such as the ligand sonic hedgehog (*shh*), the receptors *patched-1* (*ptch1*) and *patched-2* (*ptch2*) and the downstream activators *gli2a* and *gli3*. We found no discernable changes in *fsd1*^{-/-} by WISH experiment, compared to controls (see figures below). Based on these data, we concluded that Hh signaling was not affected in *fsd1*^{-/-}, therefore, Hh signaling is not involved in *fsd1*-mediated signaling and function.

4. The authors previously showed that *klf2a* is necessary for HSC generation. Even though *klf2a* function has recently been challenged in HSC generation (see Novodvorsky P et al., PLoS One, 2015), it is surprising that the authors do not analyse *klf2a* expression in this context.

Response 4: Since the blood flow was normal in *fsd1*-deficient embryos and *klf2a* is an immediate mediator of hemodynamic forces by blood flow, we didn't examine the role of *klf2a* in this context initially. Interestingly, we indeed observed the decreased *klf2a* expression in *fsd1* morphants using qPCR, WISH and western blotting (shown below). It is possible that *klf2a* is involved in cilia regulation of HSPC development. From our results, we reasoned that *klf2a* may acts downstream of cilia to regulate HSC generation. However, the exact mechanism remains unclear and awaits further investigation. In Peter Novodvorsky's paper⁵, they observed no HSC defects in *klf2*^{ash317} mutant, and one possible explanation for the discrepancy between morphants and mutants is that there may be redundancy between *klf* family members in the genetic mutation of *klf2a*.

5. The authors (and others) have shown that the EdCs are ciliated, but they have not shown that HSCs are ciliated. This needs to be clarified experimentally by performing detailed analysis of cilia

on HSCs.

Response 5: Through live imaging of Tg(*kdr1*:mCherry/ β act:Arl13b-GFP/*runx1*:en-GFP) from 25 hpf to 40 hpf, we observed that the ECs (*kdr1*⁺) and HE (*kdr1*⁺ *runx1*⁺) cells are ciliated (please see **Fig. 1A and 1B**), while the HSPCs (*kdr1* *runx1*⁺) are not (please see the figure below, green-only cell). To further confirm the HE cells are ciliated cells, we used the Tg (*gfi1*:GFP)⁶ outcrossed with Tg(β act:Arl13b-GFP) and then visualized the GFP⁺ cells at 28hpf. We used two different methods to visualize the GFP+ HE cells. One is direct live imaging of the ventral wall of DA in Tg (*gfi1*:GFP/ β act:Arl13b-GFP) embryos by confocal microscopy, and the other is fluorescence activated cell sorting of dissected trunk region of Tg (*gfi1*:GFP/ β act:Arl13b-GFP) embryos. We can see that the GFP⁺ HE cells are ciliated (**Fig. 1B**, upper and lower panels). Furthermore, to confirm that the HSPCs are not ciliated, we used the *cmvb*⁺:GFP to mark HSPCs and performed FISH (fluorescence in situ hybridization; the expression of *cmvb* probe was tested in Tg(*cmvb*:EGFP) embryos and Ac-tubulin staining in Red to label cilia). The *cmvb*⁺ HSPC was not co-stained with Ac-tubulin marked cilia (**Fig. 1C**). Therefore, we could conclude that the ECs and HE cells are ciliated, whereas the HSPCs are not.

6. A related concern, the authors mentioned that notch signaling act downstream of cilia, implying that cilia function cell-autonomously to regulate notch signaling. However, this is not assessed experimentally. This point is key for the conclusions and the authors should assess if cilia function cell autonomously or non-cell autonomously.

Response 6: Thank you for raising this important issue. We found that the target genes of Notch signaling were decreased in sorted *fli1a*⁺ vascular ECs of *fsd1* morphants, compared to controls (see figure below). This suggests that the Notch signaling was specifically decreased in vascular ECs of *fsd1* morphants. Furthermore, we performed rescue experiments with overexpression of *fsd1* or *NICD* specifically in vascular ECs by injecting *fli1a:fsd1* or *fli1a:NICD* together with Tol2 mRNA. We found that the reduced *runx1* expression in *fsd1* morphants was efficiently rescued (**Supplemental Fig. 4B and 4C; Fig. 7A-C**). Taken together, these data support that the cilia modulate HSPC development through Notch signaling, in a cell-autonomous manner.

7. In the in situ data, the sample number is low even though the authors use MO which usually allows to obtain lots of embryos for the analysis. In that aspect, one expect the phenotype to fluctuate from embryo to embryo but the authors never mention any classification of the phenotypes and their variability? For example, in Figure 3A, which signal is pointed out by the red arrowhead? It is very unclear.

Response 7: Thank you for this important suggestion. The in situ data shown in previous figures were chosen from one representative experiment. The total number of embryos used in the in situ experiments were summarized in Supplemental data 2. For example, in Fig. 3A, the *runx1* expression was summarized (please see the following figure A) and the total number was added in the revised Fig. 3A.

Some details in our WISH experiments: we usually use 10-30 embryos/probe to detect gene expression in one experiment, and then image 2 or 3 representative embryos in the same group. The phenotype of low frequency of embryos was regarded as variation. For example, in Fig. 3A, the expression of *runx1* in control embryos, 2 embryos were imaged (see the following figure B). “14/16” represents 14 out of 16 embryos showing the similar expression pattern as the figure presented. The remaining 2 embryos were regarded as variation (2 out of 16). The dashed line in the figure below shows the vessel region in the AGM region (red dashed lines, DA represents dorsal aorta and CV labels cardinal vein). And the *runx1* signals (purple signals) were calculated by the specific expression in the ventral wall of DA in the AGM region. All the in situ data were analyzed based on the indicated expression in the corresponding region. We have added the number of embryos used in the in situ experiments in the revised figures. A revised and clear description was also added in the revised manuscript.

8. The authors performed injection the *fli1a:ift88-cKO* DNA. This transient knock-down approach

is not completely appropriate to describe the *fli1a:ift88-cKO* in the Figure 4. Rescuing the *ift88* mutants with *fli:ift88* is a much better way to assess the tissue specificity of *ift88* function. In addition, considering *ift88* has cilia independent functions, *kif3a* or any other tissue specific rescue of the cilia mutants would be required.

Response 8: We thank the reviewer for pointing out this rescue experiments. We tried to generate the plasmid *fli1:ift88-EGFP* for overexpression, but failed. Therefore, we provided the conditional KO data instead.

As stated in **Response 1**, we have added more data from other cilia mutants including *pkd2*^{-/-}, *kif3a*^{-/-} and *ift88*^{-/-} mutants to support our conclusion.

Reviewer #2 (Remarks to the Author):

In this study, Liu et al show that embryonic endothelial cells have cilia which is important for the formation of hematopoietic progenitors in the zebrafish embryo. By using morpholinos and CRISPR technologies they disrupt different genes involved in cilia formation and report a hematopoietic phenotype. They further study the effect on hemogenic endothelium (HE) cells by using transgenic reporters (*runx1*, *kdr* or *c-myb*, *kdr*). They report a specific requirement of cilia in the formation of HE and in Notch activation.

Although the findings are interesting and novel, this work has several fundamental problems that would require a big amount of experiments to resolve and the current conclusion may not stand.

The main concern is that the genetic mutants they analyze in most of the experiments are general mutants that may affect other cell types and other decisions. The phenotype on *Runx1* cells that the authors observe after morpholino treatment is only happening after certain stage (26hpf) and while many determinations are performed at 30 to 36 hpf, many vascular controls and other tissue integrity are performed at 26-28 hpf.

The specific comments for each figure are:

Figure 1

What is *Arl13b*? Is that a single cilium per cell what they show in the image and estimate from quantification? Could they give some information on that?

The authors performed the first analysis with a *Runx1* reporter, are all endothelial cells that they find with cilia negative for *Runx1*? That does not agree with the next figures, could they show *runx1*⁺ cells with cilia?

Response 9: We thank this reviewer for pointing out this important issue.

1) *Arl13b* is a member of Arf/Arl subfamily of small GTPases and localized in cilia⁷. It was reported that *Arl13b* is required for ciliogenesis

2) Yes, it is. There is a primary cilium per endothelial cell, which had been reported before⁸.

3) We are sorry for the confusion. The circulating *runx1*⁺ cells in the blood vessel at 28 hpf are most erythrocytes and *runx1*⁺ cells in the ventral wall of dorsal aorta are HE cells and

emerging HSPCs⁹. The probability of ECs becoming HE cells is very low, so in this first analysis we only showed the existence of cilia in ECs. Meanwhile, we did observe cilia in *runx1⁺/kdr1⁺* cells (HE cells), but not in the nascent *runx1⁺/kdr1⁻* cells (HSPCs). Please see the revised Fig.1A and 1B, as well as the following figure (Also see **Response 5** above). Furthermore, the *cmvb* labeled HSPCs was also non-ciliated (see **Fig. 1C**).

Collectively, the HE cells can produce cilia, while the nascent HSPCs can't.

Figure 2

What is the reference for the 'newly identified centrosomal gene *fsd1*'. Could they show staining for *fsd1* and the other cilia genes used?

As previously said, the phenotype of morpholinos is very broad, in Figure 2B the endothelial layer is disrupted the defects could come from cilia or from something else.

Response 10: Thank you for raising this critical point.

1) FSD1 localization in cilia was based on the co-staining of FSD1 with Ac-tubulin by immunofluorescence (**Supplemental Fig. 1A**). In addition, we also demonstrated that knockdown of *FSD1* by using *siFSD1* in RPE-1 cells, disrupted the ciliogenesis (see **Supplemental Fig. 1A**).

2) In general, the vasculature in *fsd1*-deficient embryos developed grossly normal (see the following new figure A; Dashed lines mark the DA region). We now replaced the original figure with a more representative one in new Fig. 2B and also the data showing four Tg(*kdr1*:mCherry/ β act:Ar113b-GFP) embryos, both in control and *fsd1* morphants (please see the following figure B). From the following figure we can see that the vasculature was grossly normal in *fsd1* morphants. At the same time, we also performed fluorescence activated cell sorting of dissected trunk region of Tg(*kdr1*:EGFP) embryos at 24 hpf (before HE specification) and 28 hpf (after HE specification) and we found no difference in the number of *kdr1⁺* ECs between control and *fsd1* morphants (R2 represent the *kdr1⁺* ECs in the trunk region of the DA, please see the following figure C and D). Taken together, these data showed that the blood vessel was normal in *fsd1* morphants (see **Supplemental Fig. 9**).

Although the gross vasculature is normal, the HE cells and ciliogenesis were impaired in *fsd1* morphants. We reason that this is due to the impairment of downstream Notch signaling. Importantly, we could rescue the HSPC defects in *fsd1* morphants by overexpression of *fsd1* or NICD specifically in ECs (see **Supplemental Fig. 4B and 4C; Fig. 7A-C**), suggesting that cilia modulate HSPC through downstream Notch signaling.

Figure 3

Figure 3supl: the population $kdr^{+}runx^{+}$ has a strange expression pattern for $flk1$ (negative?) while expressing a lot of $c-myb$. Similarly in $kdr^{-}runx^{+}$ there seem to have very little $c-myb$ expression. How were these populations isolated? How many replicates were done? Were they independent biological replicates?

Surprisingly the authors claim that $kdr^{+}runx^{+}$ contain cilia but cells in Figure 1 were $runx1$ negative or at least they never show that cilia is on the HE cells.

The authors show that there are less $runx1^{+}$ and $cmyb^{+}$ cells in the morpholino treated embryos, however they should show how does the general vasculature looks like in these mutants at the same time of the analysis.

Response 11: We thank this reviewer for pointing out these critical points.

1) We are very sorry for using the wrong figure showing the expression of $cmyb$, which is now replaced by the correct one (see revised **Supplemental Fig. 3A**). The figure showed the relative expression of kdr and $cmyb$ in $kdr^{+}runx1^{-}$ ECs, $kdr^{+}runx1^{+}$ HE cells and $kdr^{-}runx1^{+}$ HSPCs, in which HE gradually lose their EC identity and gain HSPC identity, therefore the expression of kdr in HE was lower than that in ECs; conversely, the $cmyb$ expression is the lowest in ECs but much higher in HE and HSPCs.

2) The cell populations were isolated from the dissected trunk region of $Tg(kdrl:mCherry/runx1:en-GFP)$ embryos at 26 hpf. Three independent biological replicates were performed.

3) In Fig. 1, we observed the existence of cilia in ECs of DA from where the HSPCs generated. Here we showed the $kdr^{+}runx1^{+}$ cells (HE cells) are also ciliated (please see **Fig. 1B**). To confirm the existence of cilia in HE cells, we used another well-known HE transgenic line, $Tg(gfi1:GFP)$. Through live imaging of $gfi1:GFP/\betaact:Arl13b-GFP^{+}$ in the ventral wall of DA in the AGM region and fluorescence activated cell sorting of dissected trunk region of

gfi1:GFP⁺ embryos at 28 hpf, we did observe *gfi1*:GFP⁺/βact:Arl13b–GFP⁺ double positive cells (please see the new **Fig. 1B**). All together, these data showed that the HE cells are ciliated.

4) The general vasculature appears normal by examining the pan-vascular marker *kdrl*, arterial markers *ephrinB2a* and *dll4*, and venous marker *flt4* at 26 hpf and 36 hpf, by whole mount in situ hybridization in *fsd1* mutants (please see figure below and **Supplemental Fig. 9B and 9F**). Meanwhile, we also examined these vascular markers in other cilia-deficient embryos and observed no obvious differences (see **Supplemental Fig. 9D-F**). Please also see **Response 10**.

Figure 4

In these experiments they KO *itf88* in *fli1* cells and find that hematopoietic cells are decreased, although it is not too obvious from the image showed in figure 4E. Why do they use *itf88* instead of *fsd1*? Is it assumed that all these genes are doing the same but they may not, so it would be more informative to use the same ones for all the studies. In figure5 they focus again in *fsd1* again.

Response 12: We thank the reviewer for pointing out the issues. *itf88* and *fsd1* are both required for ciliogenesis¹. Furthermore, knockdown of *itf88* or *fsd1* led to defects in cilia number and length (**Fig. 2B-F**), which implying *itf88* and *fsd1* may play similar functional role in regulating ciliogenesis. In addition, the cilia number and length in DA influence HSPC development. Therefore we chose *fli1:itf88*-cKO to detect the HSPC phenotype. We agree with the reviewer on arguing “it is assumed that *itf88* and *fsd1* are doing the same but they may not”. So we tried more than three times to construct a *fli1a:fsd1*-cKO plasmid, but failed, therefore we used *itf88* as a powerful ciliogenesis-regulating gene to study the tissue-specific KO experiments.

Furthermore, we performed rescue experiments with overexpression of *fsd1* specifically in vascular ECs by injecting *fli1a:fsd1* together with Tol2 mRNA. We found that the reduced *runx1* expression in *fsd1* morphants was efficiently rescued (**Supplemental Fig. 4B and 4C**).

Figure 5

Authors check expression of EphrinB2 and Dll4 in the morphants and find that vasculature is normal. However they check at 24 hpf, also at 26hpf, but most of their analysis are made at 31-32-36 hpf. Is the vasculature and arterial/venous makers still normal? Is the skeletal muscle still normal? Some images (Fig2) don't look so normal.

Response 13: Thank you for raising this critical point. We examined the expression of pan-vascular marker *kdrl*, arterial marker *ephrinB2a*, venous marker *flt4* and skeletal muscle marker *myod* at 31 hpf, 32 hpf and 36 hpf by WISH in *fsd1/ift88/pkd2/kif3a* morphants and we did not observe significant change between control and morphants (please see **Supplemental Fig. 9F**). Therefore, the vasculature and skeletal muscle are grossly normal in cilia defective morphants.

We apologized that we used the wrong vascular image in Fig. 2 in our first submission (we did inform the handling editor about this mistake immediately after submission, but unfortunately this was not corrected in the submitted files), which is now replaced by a representative one (please also refer to **Response 10**).

Figure 6

They perform RNA-seq experiments by dissecting the trunk of the embryo of control and *fsd1* morphants. This strategy has the problem that many cell types were sequenced and it is unlikely that it represents the endothelium or HE population.

Next they perform a WB with NIIC and find that active Notch1 has almost disappeared, but the quality of this blot is not great and it is surprising that it is abolished. Such abolishment of Notch activity should result in the loss of the arterial program, which at least at 26 hours was not lost, maybe it is lost at this time.

By the *tp1* reporter they also show reduction of Notch activity, but not a complete abolishment. These data together shows some contradictions that are difficult to reconcile.

It is true that they rescue the *runx1* expression in the mutants with the overactivation of Notch, but this is not too surprising since Notch can induce *runx1* in different types of endothelial cells in the fish, so it may just be overriding any previous defect.

In general the authors should be more consistent in the time of analysis that they use and should specifically knock out the genes in the endothelium to study the effect on HE. The transcriptome and Notch studies should also be more specific.

Response 14: We thank this reviewer for pointing out these important issues.

1) We agreed that the RNA-seq performed with trunk tissues can't represent the vascular ECs or HE cells. Given that, we performed cell sorting and real-time PCR for further validating the potential targets, which were screened out by RNA-seq. We injected the *fsd1* aMO into the Tg(*fli1a*:EGFP) embryos and then performed fluorescence activated cell sorting of dissected trunk region of *fli1a*:EGFP⁺ embryos to sort *fli1a*:GFP⁺ vascular ECs in both control and *fsd1* morphants at 26 hpf. Through qPCR analysis, we found that the Notch target genes were

decreased in *fli1a*⁺ ECs of *fsd1* morphants, compared to controls (please see figure below). Notch signaling was specifically down regulated in vascular ECs in *fsd1* morphants, suggesting that the Notch signaling lies down stream of *fsd1* in ECs.

2) Furthermore, overexpression of NICD specifically in vascular ECs could rescue the HSPC defects in *fsd1* morphants at both 26- and 36 hpf (please see **Fig. 7A-C**). Altogether, these data showed that the *fsd1* regulates cilia-Notch in a cell autonomous manner.

3) We also examined the Notch signaling in *fsd1*^{-/-} mutants at 26 hpf (please see **Fig. 6C and 6D**). The decreased Notch target genes were observed in *fsd1*^{-/-}. The NICD expression at protein level was decreased in dissected trunk region of *fsd1*^{-/-} embryos, compared to wildtype embryos. Altogether, these data showed that the Notch signaling was disrupted in vascular ECs of *fsd1*-deficient embryos.

Reviewer #3 (Remarks to the Author):

COMMENTS FOR AUTHORS

In this manuscript, Liu et al present experimental studies implicating cilia, via a contribution to notch signalling, as important for hematopoietic stem cell specification in the hemogenic endothelium. I really enjoyed reading this manuscript. It is very well presented and the studies very comprehensively test the hypothesis. The experimental strategy generally presents multiple lines of evidence, often that is technically independent, to support each point.

MAJOR COMMENTS

1. The studies are comprehensive, but exclusively conducted in zebrafish. For the generic claim of the title to be true, some evidence that the pathway is conserved is required i.e. that it applies in mammalian systems also. Alternatively, the title would be acceptable without further experiments if it restricted the claim to zebrafish.

Response 15: Thank you for your suggestion. We have observed that the FSD1 protein localizes in cilia and the Fsd1-knock down led to reduced ciliogenesis and reduced nuclear NICD in human RPE-1 cells (please see the revised **Supplemental Fig. 1A** and **Fig. 6H-K**). Therefore, the mechanism of *fsd1* mediating cilia-Notch signaling appears conserved from zebrafish to mammalian systems (human RPE cells).

2. A role for Notch signalling in HSC specification is well established, including in zebrafish experimental systems, and so the involvement of Notch signalling in HSC specification per se is not a discovery of this report. Nor is the localisation of Notch pathway components and cilia (as acknowledged in citations 24 and 25). So it is the linking of these two elements that is the novelty. The introduction and discussion would contextualise the contribution of this report better if it cited some of the Notch-HSC specification literature, particularly the zebrafish contributions (e.g. Berns et al, *Genes Dev* 2005; Bertrand et al, *Blood* 2010; Clements et al *Nature* 2011; and a review, Butko et al *Dev Biol* 2016). In fact, the present studies rescue HSC emergence in the absence of cilia by overexpressing Notch - that is, presuming that cilia presence is not itself restored by enforced Notch expression. Whether this is occurring or not is not experimentally addressed. If it is not, the hemogenic endothelial cilia may be the usual site of the Notch contribution, but are not the only possible site.

Response 16: We thank the reviewer for the thoughtful suggestions. We have added the related Notch-HSPC specification literatures in the revised manuscript (please see paragraph 3 of the introduction part).

That overexpression of NICD in cilia-deficient embryos could rescue HSPC defects suggests that 1) Notch acts downstream of hemogenic endothelial cilia to regulate HSPC emergence, and 2) in the absence of cilia, ectopic intracellular Notch signaling can bypass the requirement of cilia transduction to facilitate HSPC emergence. However, we cannot exclude a possible source of Notch signaling from other non-ciliated cells.

3. The *fli1a:ift88-cKO* construct is a good experiment, achieving tissue-specific CRISPR/cas9 tissue-specific knockdown, of which there are only a few examples to my knowledge in zebrafish. 3a. Insufficient methodological detail is provided for others to replicate this experiment (sFig5 is not enough). What form of cas9 was used? What was the sequence of the gRNA in the transgene? Which U6 promoter was used? When the genotyping primers for which the sequences are provided are used, what is the distinguishing outcome?

Response 17: Thank you for pointing out this issue.

The detailed method to generate tissue-specific knock-down vector has been illustrated previously¹⁰. In detail, the *fli1a:ift88-cKO* construct for endothelial-specific gene targeting was generated by cloning the *fli1a* promoter into the pDestTol2pA2-U6:gRNA (guided RNA) (Addgene #63157) using the gateway system. Then the guide RNA of *ift88* was inserted into the above plasmid to generate the final construct for injection. The validation of *fli1a:ift88-cKO* was reported by Massimo M. Santoro (University of Turin, Turin 10126, Italy)¹¹.

1) The *Streptococcus pyogenes* CRISPR/Cas9 system was used here (Addgene #63157)^{10, 12}.

2) The *ift88* gRNA target sequence is 5'-GGCTGACCGCTATGCAGAGC-3'¹¹.

3) The U6-3 promoter was used, and the sequence of U6-3 promoter is listed in the Addgene #63157.

4) We used T7E1 enzyme digestion to distinguish mutants from control (please see the figure

below, the *cmlc2*⁺ cKO group showed two bands while *cmlc2*⁻ control showed one band). The primers for genotyping and CRISPR plasmid construction were listed in Supplemental tables S5. The embryos that showed expression of EGFP in the heart were collected for phenotype and genotyping analysis, while the embryos showing no expression of EGFP were used as controls.

3b. The label sFig5C was initially confusing for me – it would have helped if it had been labelled to show that the genotype was Tg(*kdrl*:mCherry/*cmyb*:EGFP)⁺ (the construct in B). Although I realise that *flk1* (used in the legend) and *kdrl* (used in the figure) are the same (by referring to ZFIN ID: ZDB-GENE- 000705-1), it is unnecessarily confusing, particularly to those not in the field, to use both interchangeably like this. Similarly, there is use of both terms in sFig3.

Response 18: We have changed the “*flk1*” to “*kdrl*” throughout in the manuscript and the proper description was added in the revised Supplemental Fig. 3A and Supplemental Fig. 6C.

4. Following on from this, in sFig3A it was surprising that the level of *flk1*(=*kdrl*) expression was so low in the *runx1*⁺*kdrl*⁺ (yellow) population. Why is this so?

Response 19: We thank this reviewer for pointing out these issues. The figure showed the relative expression of *kdrl*, and that HE cells gradually lose their EC identity and gain HSPC identity, so the expression of *kdrl* in *kdrl*⁺*runx1*⁺ was lower than that in *kdrl*⁺*runx1*⁻ cells. Please also see **Response 11** (point 1).

OTHER COMMENTS

1. The Actin control for the Western blots in Fig 3 and SFig are, I think, the same image. While this is possible (both show control 36h and *fsd1* aMO lanes), this should be explicitly stated, particularly as the actin control in Fig 6E (which also has the same lanes) is not.

Response 20: We thank this reviewer for pointing out this important issue. We have added a new Fig. 3B and Supplemental Fig. 1B in the revised manuscript.

2. Descriptive statistics - It was often difficult to be sure of n-values, or what n-values meant, for the descriptive statistics.

2a. Fig 1C, D. No n values provided.

2b. For all other instances of graphs with y-axes labelled “The length of cilia in blood vessels (microm), what is the n-value? For example, Fig 2F, it appears to be only n=4-6/group, which seems like too small a sample given the variation in cilial length shown for this variable in the adjacent Fig 2D – despite the statistical significance, the small n-values provide too much opportunity for biased selection.

Response 21: We have revised this part accordingly.

1) We have added the n values in the revised manuscript.

2) We have revised this part with the right y-axes in the revised manuscript. To avoid biased selection, the number of embryos in *pkd2/kif3a/ift88* morphants was enlarged (please see new **Fig. 2E and 2F**) and the n values were added in the revised manuscript (please see the figure legends in the revised **Fig. 1E and 1F; Fig. 2E and 2F**).

3. Statistics- replication.

3a. While many observations are supported by multiple lines of evidence, and are hence independently corroborated, it is often unclear how many times an individual line of evidence was replicated. For example, in Fig 4e, there are 7 data points per group – were these all obtained from one experiment in one day, which would be one experiment with n=7. Also, in this case, are the test data points unselected as the legend suggests by saying “*fli1a:ift88*-cKO-injected embryos”, or were they selected for expression of the *cmlc2*-EGFP backbone marker in the transgene, which would make sense? The same query applies to Fig 4B and C, Fig 6I and in many Supplemental figures.

3b. Fig 5J and K. These appear to be replicate experiments – was this intended? If so, the reason for presenting two experiments in this instance is not stated.

Response 22: We apologize that we didn't provide clearer statistics data.

1) We have reanalyzed these statistics and replicates (including main Figures and Supplemental Figures) and added the information in the revised figures, corresponding figure legends and supplemental data.

2) *fli1a:ift88*-cKO-injected embryos are *Cmlc2*:EGFP-positive and *Cmlc2*:EGFP-negative embryos. We also pointed out that “*Cmlc2*:EGFP-negative embryos were used as a negative control (control)” in the figure legend (Fig.4E).

3) Fig 5J (revised **Fig. 5K**) and 5K (revised **Fig.5L**) represent different stages (26 hpf and 28 hpf) of cilia in the DA in the AGM region.

4. Statistics – other questions

a. Were the t-tests two-tailed or not (I see no reason why they should not have been 2-tailed).

b. In the instance where multiple pair-wise comparisons are being performed by multiple T-tests (as in sFig10B, but also elsewhere), an adjustment for multiple comparisons is required.

Response 23: The student *t*-tests were two-tailed. Meanwhile, the multiple pair-wise comparisons were also performed by one way-ANOVA or two way-ANOVA through using GraphPad Prism 6.01 software in the revised manuscript.

Editing.

1. Sometimes the y-axis label is provided as a title to a figure panel rather than adjacent the y-axis (Fig 1C,D; 4E (in contrast to B and C); Fig 5 D,F (in contrast to J, K); sFig 4C). I suggest consistent use as a y-axis label.

2. sFig1B. The axis label states “%” but the axis range is 0-1.5; either change the numbers to %s or use the term “proportion”.

3. The English is excellent and requires very little editing.

Line 35 – delete “of”

Line 46 – add “a” following “led to”

Line 57 – last word “of” instead of “on”

Line 98 – space between “region in”

Line 105 – delete “,” after fsd1

Line 133 – Greek letter “mu” for “sum”, not letter “U”

Line 170 – suggest “achieved” rather than “employed”

Response 24: We thank this reviewer very much for listing out all these editing issues. We have revised all of the above mentioned points in the revised manuscript to ensure accuracy and clarity.

References

1. Tsujikawa, M. & Malicki, J. Intraflagellar Transport Genes Are Essential for Differentiation and Survival of Vertebrate Sensory Neurons. *Neuron* **42**, 703-716 (2004).
2. Bisgrove, B.W., Snarr, B.S., Emrazian, A. & Yost, H.J. Polaris and Polycystin-2 in dorsal forerunner cells and Kupffer's vesicle are required for specification of the zebrafish left-right axis. *Developmental biology* **287**, 274-288 (2005).
3. Sun, Z. *et al.* A genetic screen in zebrafish identifies cilia genes as a principal cause of cystic kidney. *Development* **131**, 4085-4093 (2004).
4. Pooranachandran, N. & Malicki, J.J. Unexpected Roles for Ciliary Kinesins and Intraflagellar Transport Proteins. *Genetics* **203**, 771-785 (2016).
5. Novodvorsky, P. *et al.* klf2ash317 Mutant Zebrafish Do Not Recapitulate Morpholino-Induced Vascular and Haematopoietic Phenotypes. *PLoS one* **10**, e0141611 (2015).
6. Wei, W. *et al.* Gfi1.1 regulates hematopoietic lineage differentiation during zebrafish embryogenesis. *Cell research* **18**, 677-685 (2008).
7. Caspary, T., Larkins, C.E. & Anderson, K.V. The Graded Response to Sonic Hedgehog Depends on Cilia Architecture. *Developmental cell* **12**, 767-778 (2007).
8. Goetz, J.G. *et al.* Endothelial cilia mediate low flow sensing during zebrafish vascular development. *Cell reports* **6**, 799-808 (2014).
9. Zhang, P. *et al.* G protein-coupled receptor 183 facilitates endothelial-to-hematopoietic transition via Notch1 inhibition. *Cell research* **25**, 1093-1107 (2015).
10. Ablain, J., Durand, E.M., Yang, S., Zhou, Y. & Zon, L.I. A CRISPR/Cas9 vector system for tissue-specific gene disruption in zebrafish. *Developmental cell* **32**, 756-764 (2015).

11. Chen, X., Gays, D., Milia, C. & Santoro, M.M. Cilia control vascular mural cell recruitment in vertebrates. *Cell reports* **18**, 1033-1047 (2017).
12. Jao, L.-E., Wente, S.R. & Chen, W. Efficient multiplex biallelic zebrafish genome editing using a CRISPR nuclease system. *Proceedings of the National Academy of Sciences* **110**, 13904 (2013).

Reviewers' comments:

Reviewer #1 (Remarks to the Author):

The authors made great improvements however there are still major concerns:

1. The guidelines for morpholino usage are still not followed.
2. There is still a concern about the cell autonomous requirement of cilia in modulating notch signaling. mRNA sequencing cannot address this question, only mosaic experiments. The authors should tone down this conclusion if they cannot perform mosaic experiments.
3. Knocking down gene with MO in a mutant background is not the best approach to erase maternal contribution. The authors should tone down these conclusions if they cannot perform maternal zygotic mutants.

Reviewer #2 (Remarks to the Author):

The authors have addressed several of the original concerns. For example, I am convinced that HE cells contain cilia and disruption interferes with the process of HE and HSPC specification. However, downstream effects of disrupting cilia are not clear and they are not specific of HE. They show how Notch family genes are downregulated in the EC, but not in HE.

Moreover, Figure 6 B: can they show where other endothelia/arterial Notch targets are? Eg. EphB2, Dll4 because maybe the Notch downregulation is only occurring in the EC. They show in suppl fig 9 that these genes are similarly expressed in the *fsd* mutants, which I find it surprising considering the downregulation of Notch in EC and the general downregulation of *hey2* (supl fig 11)

'Through qPCR analysis, we found that the Notch target genes were decreased in *fli1a*+ ECs of *fsd1* morphants, compared to controls (please see figure below).'

The authors should check HE cells.

'The NICD expression at protein level was decreased in dissected trunk region of *fsd1*^{-/-} embryos, compared to wildtype embryos.'

Can they show other regions of the embryo? It may just happen all over the endothelium. Staining with N1IC would be good. Does the active N1 ab work in zebrafish (that would be even better). The WB lanes for the *fsd* mutant seem to have less protein, although b-actin is similar (supl fig 6D), however b-actin seems to be saturated to see differences, do they have a lower exposure of the b-actin blot?

The authors have performed some experiments with RPE-1 cell line (KD *fsd1*) to claim that the Notch-cilia dependent activation is a general mechanism (as stated in the title). However, the title is not on the general connection of Notch and cilia, but refers to the general contribution of cilia to HSC specification through Notch but they do not demonstrate that. The title should be restricted to cilia effects and to zebrafish, which is of interest too.

Reviewer #3 (Remarks to the Author):

I have re-read the revised manuscript, the other two reviews and the author responses to those, and particularly focussed on the authors' responses to my previous comments.

This is a very substantial body of well-conducted hypothesis-based experimentation, all presented in an exceptionally clearly written manuscript. The author responses and revisions have adequately addressed most of my previous comments, with the following exception:

Comment 1 / Response 15.

The new experiment in human hTERT-RPE-1 cells (an immortalized retinal pigment epithelial cell line) establishes that the *fsd1>cilia>Notch* signalling pathway is conserved in both zebrafish and humans, but it does not experimentally establish that it is conserved in “regulat[ing] hematopoietic stem and progenitor cell specification”, which is what the title states. For the title to accurately summarize what the data in the paper show, it need the words “ in zebrafish” at the end.

Minor things:

Line 366 – I think “Santa” should be “Santa Cruz Biotechnology”

Figure 7C – y-axis legend “cells” not “cels”

Table S3 – the *her9* line is in a different font

Point-by-Point Response to Reviewers' comments

We thank the editor and three reviewers for their time and efforts dedicated to provide insightful feedback to our first revision. Here, we have added more experiments, such as strictly following the guidelines of morpholino usage and tissue specific analysis of Notch signaling. The detailed point-by-point responses to reviewers' comments are shown below.

Reviewer #1 (Remarks to the Author):

The authors made great improvements however there are still major concerns:

1. The guidelines for morpholino usage are still not followed.

Response 1: We thank this reviewer for pointing out this critical point. According to the morpholino (MO) usage guidelines, we performed several experiments to validate MOs, including second MO injection, phenotypic rescue experiments and mutant generation.

(1) For *fsdl* MO, we used two different MOs (atgMO and splice MO) to validate the defects in ciliogenesis and hematopoietic stem and progenitor cell (HSPC) development (**Fig. 2B-D, supplemental Fig 2A-F, supplemental Fig. 3B, Fig. 3A and 3B, Fig. 5A and 5B, Fig. 5D-G, supplemental Fig. 8**). Then rescue experiments by injection of *fsdl* mRNA lacking the MO-binding site demonstrated that the HSPC defects in *fsdl*-deficient embryos could be rescued (**supplemental Fig. 4A-C**). And we also observed consistent HSPC defects in *fsdl* maternal-zygotic mutants (**Fig. 3C and 3D, Fig.5C**).

(2) The *pkd2/kif3a/ift88* MOs used here has already been reported in previous papers (Pooranachandran and Malicki, 2016; Kramer-Zucker et al., 2005; Sun et al., 2004). We could mimic the phenotypes reported in these studies, such as curved body (**supplemental Fig. 1J and 1H**). And disrupted ciliogenesis and HSPC development were dose-dependent (please see the **Response Fig. 1A-1C**). In addition, the corresponding atg-MO-resistant form of *ift88/kif3a* mRNA could rescue the ciliogenesis and HSPC defects in morphants, respectively (please see the **Response Fig. 1B and 1C**).

Taken together, we have now strictly followed the MO guidelines and the detailed experiments that we had performed were summarized in the following table.

Response figure 1

Guidelines for MOs	fsd1	pkd2/kif3a/ift88
MO efficiency	fsd1 aMO was efficient. (western blot; sFig. 1B) fsd1 sMO was efficient. (RT-PCR; sFig. 1C)	pkd2/kif3a/ift88 atg MOs were efficient. (pEGFP-N1 fused plasmids and curved body phenotypes; sFig. 1D, 1G, 1H)
multiple MOs	1. Ciliogenesis defect occurred in fsd1 aMO/sMO-injected embryos; (Fig. 2B-2D, sFig. 2A-F.) 2. HE and HSPC defects occurred in fsd1 aMO/sMO-injected embryos (sFig. 3B, Fig. 3A and 3B, Fig. 5A and 5B, Fig. 5D-G, sFig. 8)	1. Ciliogenesis defect occurred in atg MOs-injected embryos (Fig 2E and 2F, response Fig. 1A) 2. HE, HSPC defects occurred in atg MOs-injected embryos (Fig 3E, Fig. 5H-L, response Fig. 1C)
mRNA rescue	fsd1 mRNA injection rescued HSPC defects (sFig. 4A-C)	kif3a/ift88 mRNAs injection rescued HSPC and ciliogenesis defects (Response Fig. 1B and 1C)

Mutant data	HE and HSPC defects occurred in fzdl maternal-zygotic mutants (Fig. 3C and 3D, Fig. 5C)	HSPC and ciliogenesis defects occurred in pkd2/ift88 mutants with depletion of maternal expression. (sFig. 5D and 5E, Fig. 3F)
---	---

2. There is still a concern about the cell autonomous requirement of cilia in modulating notch signaling. mRNA sequencing cannot address this question, only mosaic experiments. The authors should tone down this conclusion if they cannot perform mosaic experiments.

Response 2: Thank you for raising this important issue.

(1) First, we full agree with you that mosaic experiments would be the best approach to demonstrate the cell-autonomy. We tried to perform transplantation, but failed to get enough successful transplanted embryos.

(2) Then, we tried to detect whether notch-activated ECs are ciliated. To answer this question, we outcrossed *arl13b:GFP* and *tp1:mCherry* transgenic lines, and Arl13b⁺ cilia could be detected in the *tp1*⁺ ECs in the ventral wall of DA, which implied that cilia may modulate Notch signaling in cell-autonomous manner (please see **Response Fig. 2** and **supplemental Fig.11A** in the revised manuscript.).

(3) Although we performed endothelial cell specific knock down (**Fig. 4A-C and 4E**) and rescue experiments (**Fig. 7**) and examined Notch signaling in HE using qPCR (**Fig. 6E**), there is no direct and definite evidence to demonstrate the cell autonomous requirement of cilia in modulating notch signaling. Therefore, we revised the sentence “The qPCR result showed that *notch1a* expression was decreased in HE cells specifically (**Fig. 6E**), indicating that HE cells were sensitive to Notch signaling cell-autonomously” in line 233/234 to “The qPCR result showed that *notch1a* expression was decreased in HE cells (**Fig. 6E**), indicating that HE cells were sensitive to Notch signaling”.

Response figure 2

3. Knocking down gene with MO in a mutant background is not the best approach to erase maternal contribution. The authors should tone down these conclusions if they cannot perform maternal zygotic mutants.

Response 3: We thank the reviewer for your suggestion. We observed HSPC defects in *fsd1* maternal-zygotic mutants (please see **Fig. 3C and 3D; Fig. 5C**). However, we could not get the *pkd2/kif3a/ift88* maternal-zygotic mutants, therefore we modified our conclusions in revised manuscript (page 6, the last-second paragraph, in red).

Reviewer #2 (Remarks to the Author):

The authors have addressed several of the original concerns. For example, I am convinced that HE cells contain cilia and disruption interferes with the process of HE and HSPC specification. However, downstream effects of disrupting cilia are not clear and they are not specific of HE. They show how Notch family genes are downregulated in the EC, but not in HE.

Response 4: Thanks for pointing out this issue. We sorted *runx1⁺/kdr1⁺* HE cells at 26 hpf and did qPCR to detect Notch target genes, e.g. *hey1*, *hey2* and *her2*. We found that the Notch target genes were decreased in the HE cells in *fsd1* morphants (**Response Fig. 3A**).

Response figure 3

Moreover, Figure 6 B: can they show where other endothelia/arterial Notch targets are? Eg. EphB2, Dll4 because maybe the Notch downregulation is only occurring in the EC. They show in suppl fig 9 that these genes are similarly expressed in the *fsd* mutants, which I find it surprising considering the downregulation of Notch in EC and the general downregulation of *hey2* (supl fig 11)

Response 5: Thanks for your suggestion.

(1) We reanalyzed the RNA-seq data and added endothelial genes *fli1a* and *kdrl* in volcano plot. The expression of endothelial genes *fli1a* and *kdrl* was relatively normal in *fsd1*-deficient embryos, consistent with the whole-mount in situ hybridization (WISH) results (**Response Fig. 3B; supplemental Fig. 9A and 9C**). Meanwhile, we also added *dll4*, *ephrinB2a*, *hey2* in volcano plot. And the expression of arterial markers *dll4* and *ephrinB2a* was not changed in *fsd1* morphants, while the expression of Notch target gene *hey2* was decreased. These results were consistent with the WISH results (please see **response Fig. 2B; supplemental Fig. 9B; supplemental Fig. 11B**).

(2) We were not the first to shown that not all of the Notch signaling components were altered in gene-deficient embryos. Previous paper also reported similar results. Please see the following table. It may be due to different upstream factors of Notch signaling or different mediators between NICD and its downstream targets. The detailed mechanism for this phenomenon needs further studies.

Mutants or morphants	Results	References
hif1a , hif2a -deficient zebrafish embryos	ephrinB2a (no change) gata2b , notch1a , notch1b (decreased)	(Gerri et al., 2018)
foxc1a/foxc1b -deficient zebrafish embryos	ephrinB2a , notch3 (no change) NICD (decreased)	(Jang et al., 2015)
tgfbr2 -deficient zebrafish embryos	ephrinB2a , dll4 , hey2 (no change) gata2b , jag1 (decreased)	(Monteiro et al., 2016)

‘Through qPCR analysis, we found that the Notch target genes were decreased in *fli1a*+ ECs of *fsd1* morphants, compared to controls (please see figure below).’

The authors should check HE cells.

Response 6: Please see the Response 4.

‘The NICD expression at protein level was decreased in dissected trunk region of *fsd1*^{-/-} embryos, compared to wildtype embryos.’ Can they show other regions of the embryo? It may just happen all over the endothelium. Staining with NIIC would be good. Does the active N1 ab work in zebrafish (that would be even better).

Response 7: Thanks for pointing out this issue. We tried to do immunofluorescence to detect the NICD expression in zebrafish embryos. Unfortunately, the NICD antibody did not work well in the zebrafish embryos (**response Fig. 3C**). Therefore, we chose to dissect the non-trunk region of the embryos and performed western blot to examine the NICD expression. The **Response Fig. 3D** showed that the expression of NICD was also decreased in the non-trunk region of *fsd1*^{-/-}, compared to controls.

The WB lanes for the *fsd* mutant seem to have less protein, although b-actin is similar (supl fig 6D), however b-actin seems to be saturated to see differences, do they have a lower exposure of the b-actin blot?

Response 8: We have checked the raw data (please see **response Fig. 3E**). We could see that the amount of Actin with different exposure time in *fsd1*^{-/-} was the same with that in wild types. And we have replaced western blot with actin loading controls in the revised **Fig. 6D, Fig. 3B, Fig. 3D** and **Supplemental Fig. 3E**.

The authors have performed some experiments with RPE-1 cell line (KD *fsd1*) to claim that the Notch-cilia dependent activation is a general mechanism (as stated in the title). However, the title is not on the general connection of Notch and cilia, but refers to the general contribution of cilia to HSC specification through Notch but they do not demonstrate that. The title should be restricted to cilia effects and to zebrafish, which is of interest too.

Response 9: Thanks for your suggestion. We had added “in zebrafish” at the end of the title.

Reviewer #3 (Remarks to the Author):

I have re-read the revised manuscript, the other two reviews and the author responses to those, and particularly focussed on the authors’ responses to my previous comments.

This is a very substantial body of well-conducted hypothesis-based experimentation, all presented in an exceptionally clearly written manuscript. The author responses and revisions have adequately addressed most of my previous comments, with the following exception:

Comment 1 / Response 15.

The new experiment in human hTERT-RPE-1 cells (an immortalized retinal pigment epithelial cell line) establishes that the *fsd1*>*cilia*>Notch signalling pathway is conserved in both zebrafish and humans, but it does not experimentally establish that it is conserved in “regulat[ing] hematopoietic stem and progenitor cell specification”, which is what the title states. For the title to accurately summarize what the data in the paper show, it need the words “ in zebrafish” at the end.

Response 10: Thank you for pointing out this important issue. We had added “in zebrafish” at the end of the title in the revised manuscript.

Minor things:

Line 366 – I think “Santa” should be “Santa Cruz Biotechnology”

Figure 7C – y-axis legend “cells” not “cels”

Table S3 – the *her9* line is in a different font

Response 11: Thanks for your kind suggestions and we had revised the main text in the revised manuscript.

References

- Gerri C, Marass M, Rossi A, Stainier DYR. Hif-1alpha and Hif-2alpha regulate hemogenic endothelium and hematopoietic stem cell formation in zebrafish[J]. *Blood*, 2018.
- Jang IH, Lu YF, Zhao L, Wenzel PL, Kume T, Datta SM, Arora N, Guiu J, Lagha M, Kim PG, Do EK, Kim JH, Schlaeger TM, Zon LI, Bigas A, Burns CE, Daley GQ. Notch1 acts via Foxc2 to promote definitive hematopoiesis via effects on hemogenic endothelium[J]. *Blood*, 2015, 125(9):1418-1426.
- Kramer-Zucker AG, Olale F, Haycraft CJ, Yoder BK, Schier AF, Drummond IA. Cilia-driven fluid flow in the zebrafish pronephros, brain and Kupffer's vesicle is required for normal organogenesis[J]. *Development*, 2005, 132(8):1907-1921.
- Monteiro R, Pinheiro P, Joseph N, Peterkin T, Koth J, Repapi E, Bonkhofer F, Kirmizitas A, Patient R. Transforming Growth Factor β Drives Hemogenic Endothelium Programming and the Transition to Hematopoietic Stem Cells[J]. *Developmental cell*, 2016, 38(4):358-370.
- Pooranachandran N, Malicki JJ. Unexpected Roles for Ciliary Kinesins and Intraflagellar Transport Proteins[J]. *Genetics*, 2016, 203(2):771-785.
- Sun Z, Amsterdam A, Pazour GJ, Cole DG, Miller MS, Hopkins N. A genetic screen in zebrafish identifies cilia genes as a principal cause of cystic kidney[J]. *Development*, 2004, 131(16):4085-4093.

REVIEWERS' COMMENTS:

Reviewer #1 (Remarks to the Author):

Regarding the tp1 expression with ciliated data why are some ciliated ECs without tp1 activation present in this figure? This needs to be quantified.

The error bars of Fig2C, 2D, 7C, S4C, S10B, and S10D look weird. The authors have described that the error bars show S.D. parameter, but this does not seem right when we plot the data generated by the authors. We suggest the authors to verify that they indeed use S.D. and not S.E.M.

The experiment shown in SFig1a and SFig1E were already done in the same group paper (Fig1a, Fig1f and g in Nat Comm, 2018). No need to include these data again.

SFig3A, they showed that the expression of *fsd1* enriched in *runx1+kdr1+* cell at 26 hpf, however, the expression of *fsd1* does not appear in EC at 26 hpf in sFig3D. Furthermore, the *fsd1* expression in this mutant (sFig3D) does not look same as the published data (shown in sFig3b of Nat Comm, 2018 and is redundant as well).

According to the MO guideline, the maternal effect of these ciliary genes should be demonstrated.

Reviewer #2 (Remarks to the Author):

One of my concerns was that the decrease in Notch-IC is not specific of HE or HSPC, but there is a general decrease in *fsd1*^{-/-}. The authors corroborate that by performing a WB of the non-trunk region, and find that it is decreased there too. However, they do not show this result nor comment in the text where they still keep pointing to a decrease in the trunk region. This can be miss-leading and they should give this information. Furthermore, they should discuss why arterial phenotype is maintained in the decreased Notch condition, while the HE are affected.

The rest of the concerns have been addressed.

REVIEWERS' COMMENTS:

Reviewer #1 (Remarks to the Author):

Regarding the tp1 expression with ciliated data why are some ciliated ECs without tp1 activation present in this figure? This needs to be quantified.

Response 1: We thank the reviewer for pointing out this issue. 1) Previous paper¹ and our data demonstrated that ECs can be ciliated, and Notch signaling was only activated in some but not all ciliated ECs. Therefore, there are some ciliated ECs without tp1 activation in our observation. 2) However, it is technically difficult to calculate the exact number of tp1-positive ciliated ECs vs tp1-negative ciliated ECs since current transgenic lines are not possible to distinguish the tp1 expression, cilia and EC simultaneously.

The error bars of Fig2C, 2D, 7C, S4C, S10B, and S10D look weird. The authors have described that the error bars show S.D. parameter, but this does not seem right when we plot the data generated by the authors. We suggest the authors to verify that they indeed use S.D. and not S.E.M.

Response 2: We apologized that we didn't provide detailed information for the error bars. Below is the detail for the error bars of all the figures in our revised manuscript.

Response Figure 2

Figure			Figure		
name	before	revised	name	before	revised
Fig.1E	SD	SD	sFig.2B	SD	SD
Fig.1F	SD	SD	sFig.2C	SD	SD
Fig.2C	SEM	SD	sFig.2E	SD	SD
Fig.2D	SEM	SD	sFig.2F	SD	SD
Fig.2E	SD	SD	sFig.4C	SEM	SD
Fig.2F	SD	SD	sFig.9A	SEM	SD
Fig.4B	SD	SD	sFig.10B	SEM	SD
Fig.4C	SD	SD	sFig.10D	SEM	SD
Fig.4E	SD	SD	sFig.11C	SD	SD
Fig.5E	SD	SD	sFig.11D	SD	SD
Fig.5G	SD	SD	sFig.11F	SD	SD
Fig.5I	SD	SD			
Fig.5K	SD	SD			
Fig.5L	SD	SD			
Fig.6G	SD	SD			
Fig.7C	SEM	SD			

The experiment shown in SFig1a and SFig1E were already done in the same group paper (Fig1a, Fig1f and g in Nat Comm, 2018). No need to include these data again.

Response 3: We thank the reviewer for this suggestion. We deleted sFig1A and sFig1E in the revised paper. Meanwhile, we also deleted the information about *fsd1* mutant that had been shown in the published Nat Comm paper (Tu et al, 2018).

SFig3A, they showed that the expression of *fsd1* enriched in *runx1+kdr1+* cell at 26 hpf, however, the expression of *fsd1* does not appear in EC at 26 hpf in sFig3D. Furthermore, the *fsd1* expression in this mutant (sFig3D) does not look same as the published data (shown in sFig3b of Nat Comm, 2018 and is redundant as well).

Response 4: Thanks for pointing out this issue. 1) In sFig3A, the qPCR result of *fsd1* in *runx1+kdr1+* showed the relative expression of *fsd1* in HE. The absolute expression level of *fsd1* in ECs is low. Through fluorescence in situ hybridization of *fsd1* in Tg (*fli1a*:EGFP) embryos at 26 hpf, we observed very low level of *fsd1* in blood vessels (please see the following Response Figure). 2) The staining intensity between these two experiments was different as the *fsd1* staining in wild type embryos in sFig3b of Nat Comm was much stronger than that in our sFig3D.

According to the MO guideline, the maternal effect of these ciliary genes should be demonstrated.

Response 5: In this manuscript, we have demonstrated that defects including cilia, left-right asymmetry and HSC can be detected in *fsd1* maternal-zygotic mutants, but not in zygotic mutants. And previous papers also showed that *ift88* (Polaris) is present maternally and most cilia are normal in *ovl^{tz288b/tz288b}* mutants but not in maternal-zygotic *ovl* mutants, because of maternal *ift88* expression^{2, 3, 4}. Since the early lethality of *pkd2^{-/-}* and *kif3a^{-/-}* zygotic mutants, we couldn't obtain the maternal-zygotic mutants at the moment. To establish such mutants will take substantial time and efforts by doing rescue experiments with mRNA injection into early zygotic mutant embryos.

Reviewer #2 (Remarks to the Author):

One of my concerns was that the decrease in Notch-IC is not specific of HE or HSPC, but there is a general decrease in *fsd1^{-/-}*. The authors corroborate that by performing a WB of the non-trunk region, and find that it is decreased there too. However, they do not show this result nor comment in the text where they still keep pointing to a decrease in the trunk region. This can be miss-leading and they should give this information. Furthermore, they should discuss why arterial phenotype is maintained in the decreased Notch condition, while the HE are affected.

The rest of the concerns have been addressed.

Response 6: Thanks for your suggestions. 1) We had added the data on the decreased Notch in the

non-trunk region in the Supplementary Figure 11C in the revised manuscript. 2) We speculate that Notch signaling is dynamically required in different processes, depending on its timing, strength/levels and other factors. For example, *ephrinB2a* and *dll4* play a role in artery formation, and our study and previous study showed that the expression of these two genes did not change, while *jagged1* is only required for HSC specification⁵. We also added this speculation in the “Discussion” section in the revised manuscript.

References

1. Chen, X., Gays, D., Milia, C. & Santoro, M.M. Cilia control vascular mural cell recruitment in vertebrates. *Cell reports* **18**, 1033-1047 (2017).
2. Bisgrove, B.W., Snarr, B.S., Emrazian, A. & Yost, H.J. Polaris and Polycystin-2 in dorsal forerunner cells and Kupffer's vesicle are required for specification of the zebrafish left–right axis. *Developmental biology* **287**, 274-288 (2005).
3. Huang, P. & Schier, A.F. Dampened Hedgehog signaling but normal Wnt signaling in zebrafish without cilia. *Development* **136**, 3089-3098 (2009).
4. Tsujikawa, M. & Malicki, J. Intraflagellar Transport Genes Are Essential for Differentiation and Survival of Vertebrate Sensory Neurons. *Neuron* **42**, 703-716 (2004).
5. Monteiro, R. *et al.* Transforming Growth Factor β Drives Hemogenic Endothelium Programming and the Transition to Hematopoietic Stem Cells. *Developmental cell* **38**, 358-370 (2016).